# Token Cleaning: Fine-Grained Data Selection for LLM Supervised Fine-Tuning

Jinlong Pang [1]  Na Di [2]  Zhaowei Zhu [3]  Jiaheng Wei [4]  Hao Cheng [5]  Chen Qian [1]  Yang Liu [1]

## Abstract

Recent studies show that in supervised fine-tuning (SFT) of large language models (LLMs), data quality matters more than quantity. While most data cleaning methods concentrate on filtering entire samples, the quality of individual tokens within a sample can vary significantly. After pre-training, even in high-quality samples, patterns or phrases that are not task-related can be redundant, uninformative, or even harmful. Continuing to fine-tune on these patterns may offer limited benefit and even degrade downstream task performance. In this paper, we investigate token quality from a noisy-label perspective and propose a generic *token cleaning* pipeline for SFT tasks. Our method filters out uninformative tokens while preserving those carrying key task-specific information. Specifically, we first evaluate token quality by examining the influence of model updates on each token, then apply a threshold-based separation. The token influence can be measured in a single pass with a fixed reference model or iteratively with self-evolving reference models. The benefits and limitations of both methods are analyzed theoretically by error upper bounds. Extensive experiments show that our framework consistently improves downstream performance. Code is available at https://github.com/UCSC-REAL/TokenCleaning.

## 1. Introduction

Supervised fine-tuning (SFT) has served as a widely adopted approach and a fundamental step in aligning large language models (LLMs) with human expectations. This process ensures that LLMs can accurately understand human instructions and produce relevant responses. In practice, SFT involves fine-tuning pre-trained models using annotated instructional data (Touvron et al., 2023). Following general data scaling laws (Zhang et al., 2024), significant efforts have been dedicated to collecting large-scale instructional data containing millions of examples (Wang et al., 2022; Chung et al., 2024; Longpre et al., 2023).

Recent studies on the SFT have widely agreed that data quality matters far more than quantity (Zhou et al., 2024; Chen et al., 2023; Pang et al., 2024b; Liu et al., 2023). That is, a small, well-curated dataset can often deliver effective or even superior performance on downstream tasks, highlighting the critical role of data cleaning or selection. Existing data cleaning approaches primarily emphasize identifying high-quality samples in large dataset pools via some metrics, including perplexity (Cao et al., 2024), completion length (Zhao et al., 2024), confidence scores (Chen & Mueller, 2024), LLM-generated quality ratings (Chen et al., 2023; Pang et al., 2024b; Liu et al., 2023) or even costly human annotations (Zhou et al., 2024). Although these methods have proven effective, focusing solely on sample-level cleaning may overlook complexities within each sample.

In practice, each sample typically contains hundreds of tokens, some of which occur frequently regardless of the sample's quality. These common tokens/patterns can overshadow task-specific words that are crucial for model performance during training. Moreover, during interference, if the model continually outputs these frequent tokens, it may neglect more informative ones, producing outputs that appear correct yet fail to address specific tasks. Thus, even well-curated samples can contain token-level noise that dilutes essential signals. For convenience, we refer to tokens that are uninformative, low-impact, or even harmful as *uninformative tokens*, and the rest as *informative tokens*. Addressing these token-level issues by removing or down-weighting the uninformative tokens can help the model prioritize important and informative tokens and then improve downstream task performance (Lin et al., 2024).

In this paper, we go beyond traditional sample-level data cleaning by proposing a generic *token cleaning* pipeline and an analytical framework for LLM SFT tasks. Specifically, we filter out uninformative tokens while retaining those with meaningful task-specific information. This is achieved by first assessing token quality using an influence-guided

---

[1]University of California, Santa Cruz [2]Northeastern University [3]Docta.ai [4]Hong Kong University of Science and Technology (Guangzhou) [5]Hong Kong Baptist University. Correspondence to: Yang Liu <yangliu@ucsc.edu>.

*Proceedings of the 42nd International Conference on Machine Learning*, Vancouver, Canada. PMLR 267, 2025. Copyright 2025 by the author(s).

scoring mechanism, followed by threshold-based separation.

Figure 1 illustrates two scoring strategies supported by our token cleaning pipeline. The high-level idea is to evaluate the influence of model updates on each token, which can be calculated by the loss disparity between a base model and a reference model. We introduce two implementations:

- *Fixed-Model Cleaning.* In this strategy, both the base model and the reference model remain fixed, and a one-shot token cleaning is applied to the entire SFT dataset. The base model is then fine-tuned on the cleaned tokens, producing the final model output. This strategy is similar to the latest token selection method for pre-trained data (Lin et al., 2024). We defer detailed comparisons to Section 4.3.1 (analytical) and Section 6 (experimental).

- *Self-Evolving Cleaning.* In this strategy, the base model remains fixed while the reference model is updated iteratively. The data is divided into multiple parts, with each iteration cleaning one part. The reference model is then updated sequentially using the cleaned results from each part. Unlike the fix-model cleaning, the final model output is the reference model obtained in the last iteration.

Our main contributions can be summarized as follows.

- *Generic Token Cleaning Pipeline.* We formulate the problem from the perspective of noisy labels and present a novel influence-guided token cleaning pipeline that scores and filters out uninformative tokens, enhancing task performance by focusing model training on the most relevant tokens. The pipeline not only encompasses existing approaches but also inspires new implementations.

- *Self-Envolving Cleaning.* Beyond merely calculating the influence scores with a pair of fixed models, we propose to update the reference model iteratively, which could progressively enhance the quality of supervision signals, leading to better downstream performance.

- *Analytical Framework.* We provide rigorous analyses to show when and why SFT with the cleaned token outperforms the full tokens. Specifically, we establish an upper bound on the error incurred when learning with full tokens (Theorem 5.1), offering theoretical insights into the trade-offs of different cleaning strategies. Our analysis explains why fixed-model cleaning yields stable but limited improvements, while self-evolving cleaning shows greater potential but requires careful implementation.

- *Comprehensive Experiments.* We conduct extensive experiments across multiple tasks, demonstrating that our token cleaning pipeline consistently boosts performance over baselines and validates its practical merits.

## 2. Related Work

**LLM Data Selection** In the LLM SFT phase, various metrics have been introduced to assess data quality including completion length (Zhao et al., 2024), perplexity (Cao et al., 2024), reward scores (Gou & Nguyen, 2024), discrete confidence scores (Chen & Mueller, 2024), the loss disparities when certain examples are included or excluded (Li et al., 2023), gradient matching (Zhou et al., 2023) and influence function scores (Xia et al., 2024). Another line of work uses advanced LLMs directly to filter out low-quality samples according to different metrics, such as quality-based rating scores (Chen et al., 2023; Liu et al., 2023; Pang et al., 2024b) and fine-grained tags (Lu et al., 2023). Diversity-aware scoring has also been integrated into the overall quality assessment, highlighting its importance. Although extensive data selection methods have shown promise, fine-grained token-level selection remains underexplored. Recent studies (Lin et al., 2024) have highlighted the significant benefits of token selection during the pre-training phase, yet its application in the SFT has received limited attention.

**Noisy Data Cleaning** The learning with noisy labels has been extensively studied (Vahdat, 2017; Veit et al., 2017; Li et al., 2017; Liu & Wang, 2021; Yuan et al., 2024). Various approaches have been proposed to mitigate label errors, including developing noise-tolerant loss functions (Natarajan et al., 2013; Reed et al., 2014; Zhu et al., 2021) and identifying clean samples while re-labeling corrupted ones (Northcutt et al., 2021; 2017; Cheng et al., 2021; Zhu et al., 2022). Recently, the issue of noisy labels in LLM alignment has gained increasing attention, driven by the observation that data quality is far more critical than quantity (Zhou et al., 2024). Recent work (Chong et al., 2022) investigated the effectiveness of leveraging pre-trained models to identify inherent label errors in natural language datasets. Additionally, efforts have been made to mitigate label errors in LLM alignment datasets (Zhu et al., 2024), particularly in the context of binary harmlessness classification. Furthermore, Pang et al. (2024b) systematically analyzed error patterns in LLM-generated quality rating scores to reduce score errors. Another line of research has focused on developing noise-tolerant DPO-based loss functions, including cDPO (Mitchell), robust-DPO (Chowdhury et al., 2024), and PerpCorrect (Kong et al.). The above studies primarily focus on noisy labels at the sample level. In contrast, our work explores fine-grained, token-level noisy labels to identify and filter out uninformative tokens, thereby boosting downstream task performance.

## 3. Preliminary

### 3.1. Next-Token Prediction

Consider a data pool comprising $N$ samples, denoted as $\{\boldsymbol{x}_i\}_{i=1}^N$. Each sample $\boldsymbol{x}_i$ represents a sequence of tokens (including the prompt and the response) defined as $\boldsymbol{x}_i := \{x_{i,j}\}_{j=1}^{L_i}$, where $L_i$ denotes the token length for the

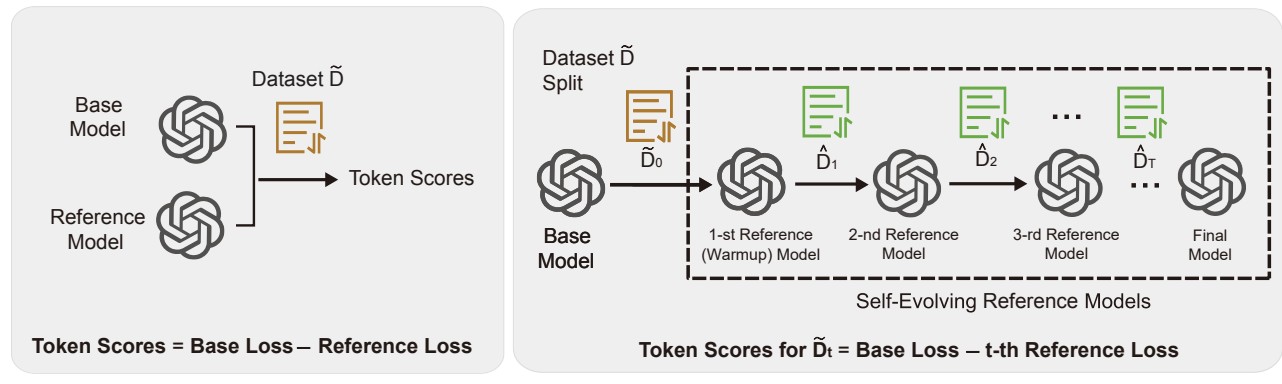

Figure 1: **Overview of the token cleaning pipeline.** *Fixed-Model Cleaning* applies a one-shot cleaning process to the entire dataset $\widetilde{D}$. In contrast, *Self-Evolving Cleaning* follows an iterative approach. It begins with a warm-up phase, where a model is fine-tuned on the full tokens of split-0, denoted as $\widetilde{D}_0$, and then used to clean the next data split, transforming $\widetilde{D}_1$ into $\widehat{D}_1$. The reference model is subsequently updated by fine-tuning the warm-up model (i.e., the first reference model) on $\widehat{D}_1$. This iterative process continues, progressively refining the reference model with each newly cleaned data split.

$i$-th sample. The training of LLMs can be framed as minimizing the negative log-likelihood of the observed tokens in the dataset. The model predicts the conditional probability $\mathbb{P}(x_{i,j}|\boldsymbol{x}_{i,:j};\theta)$ for each token $x_{i,j}$ given its preceding context, where $\theta$ represents the model parameters, and $\boldsymbol{x}_{i,:j}$ denotes the first $j-1$ tokens, i.e., $\{x_{i,1},\cdots,x_{i,j-1}\}$. Denote by $D := \{(x_{i,j},\boldsymbol{x}_{i,:j},y_{i,j}),\forall(i,j)\in S\}$, where $S := \{(i,j)|i\in[N],j\in[L_i]\},[N] := \{1,2,\cdots,N\}$. The loss function for the dataset can be expressed as:

$$\widehat{\mathcal{L}}_D(\theta) = \frac{1}{\sum_{(i,j)\in S} y_{i,j}} \sum_{(i,j)\in S} y_{i,j}\ell(x_{i,j}|\boldsymbol{x}_{i,:j};\theta), \quad (1)$$

where $\ell(x_{i,j}|\boldsymbol{x}_{i,:j};\theta) := -\log\mathbb{P}(x_{i,j}|\boldsymbol{x}_{i,:j};\theta)$, and $y_{i,j}\in\{0,1\}$ is a binary (ground-truth) label indicating whether the token $x_{i,j}$ is a valid target or not. By iteratively updating $\theta$, the model learns to assign higher probabilities to the correct tokens while disregarding irrelevant ones.

### 3.2. Token-Level Labels

Token-level labels $y_{i,j}\in\{0,1\}$ play a crucial role in determining which tokens contribute to the loss calculation. However, its ground-truth value is often unknown. Denote $\tilde{y}_{i,j}$ by the (noisy) token label that we use in practice, which may or may not be identical to $y_{i,j}$. During different training phases, the criteria for setting $\tilde{y}_{i,j}$ may vary:

- **Model Pretraining**: When training on general text data without explicit distinction between prompts and responses, all tokens are typically considered valid targets ($\tilde{y}_{i,j} = 1$), unless specific tokens are identified as irrelevant or redundant (Lin et al., 2024).

- **Supervised Fine-tuning (SFT)**: In this phase, the tokens

corresponding to the prompt part are ignored, as they do not represent the model's predictions. Therefore, for prompt tokens, $\tilde{y}_{i,j} = 0$, and for response tokens, $\tilde{y}_{i,j} = 1$.

## 4. Token Cleaning: A Noisy Label Perspective

### 4.1. Intuition

In the phase of SFT, some tokens are deemed *uninformative* or *noisy* since most of the knowledge has been obtained in the pretraining phase, e.g., common patterns and structures, high-frequency phrases. In practice, the SFT phase assigns a label of 1 to every token in the response, resulting in **noisy token labels**, where irrelevant tokens are incorrectly labeled as important ($\tilde{y}_{i,j} = 1$). Such noise can hinder the model's optimization process by introducing misleading gradients, reducing the signal-to-noise ratio by hiding informative tokens, and potentially leading to suboptimal performance.

To address this issue, it is essential to perform *fine-grained token label cleaning*. This involves identifying and filtering out uninformative tokens while preserving the tokens that carry valuable task-specific information. As suggested by the noisy label literature (Zhu et al., 2022; 2024), the cleaning process typically involves two key components: a scoring function to assess the quality of each token and a threshold to distinguish between informative and uninformative tokens, which will be detailed in the next subsection.

### 4.2. Token Cleaning Pipeline

In this section, we will introduce the main components of the token-cleaning pipeline, including the scoring function and a simple yet effective threshold.

### 4.2.1. Score Functions: An Influence-Guided Approach

We deliver our intuition when designing score functions using the following example. Suppose that the model is improved from $\theta$ to $\theta'$ by fine-tuning it on some data. According to Koh & Liang (2017); Pang et al. (2024a), the model update influences the prediction accuracy of each token, which can be written as

$$\mathsf{Infl}\,(x_{i,j}|\boldsymbol{x}_{i,:j};\theta,\theta') := \ell(x_{i,j}|\boldsymbol{x}_{i,:j};\theta') - \ell(x_{i,j}|\boldsymbol{x}_{i,:j};\theta). \tag{2}$$

Intuitively, a more negative $\mathsf{Infl}\,(x_{i,j}|\boldsymbol{x}_{i,:j};\theta,\theta')$ indicates a higher confidence improvement on predicting $x_{i,j}$ given $\boldsymbol{x}_{i,:j}$. The equation can be explained from two perspectives:

- **Assume Token Quality:** As demonstrated by Pang et al. (2024a), if we believe that token $x_{i,j}$ is the best choice given context $\boldsymbol{x}_{i,:j}$, the above influence can be used to evaluate the quality of data that brings the model from $\theta$ to $\theta'$ on this specific task, i.e, a more negative $\mathsf{Infl}\,(x_{i,j}|\boldsymbol{x}_{i,:j};\theta,\theta')$ indicates a higher data quality.

- **Assume Model Quality:** From another perspective, if we believe the model $\theta'$ performs better on $\theta$ on this specific task, the above influence can be used to evaluate the quality of token $x_{i,j}$ since a good and underfitted choice of $x_{i,j}$ tends to have a negative $\mathsf{Infl}\,(x_{i,j}|\boldsymbol{x}_{i,:j};\theta,\theta')$.

In this paper, we *assume the model quality* and use the negative of the influence defined in Eq. (2) to evaluate the quality of tokens, i.e.,

$$\mathsf{Score}(x_{i,j}|\boldsymbol{x}_{i,:j};\theta,\theta') = -\mathsf{Infl}\,(x_{i,j}|\boldsymbol{x}_{i,:j};\theta,\theta'), \tag{3}$$

where a higher score indicates a higher token quality.

Extending from the current use of influences (Koh & Liang, 2017; Pang et al., 2024a), we notice that $\theta$ and $\theta'$ are not necessarily to be the same model structure, as long as they share the same tokenizer. We will discuss potential choices of $\theta$ and $\theta'$ in Section 4.3.

### 4.2.2. Threshold

After computing token scores, a threshold is directly applied to filter out uninformative tokens. The threshold separates the tokens that significantly improve the model performance from those that do not. An ideal approach is to use algorithms to estimate the ratio of the corrupted and then select the informative tokens according to the ratio. However, although there are lots of trials in the literature on noisy labels, most works focus on cleaning the image labels (Lad & Mueller, 2023) or sample-level text labels (Zhu et al., 2024; Pang et al., 2024b). To the best of our knowledge, a feasible algorithm for estimating the noise ratio of token

labels is unclear, which is beyond the scope of our paper and left for future explorations. In this paper, we use a fixed ratio (i.e., selected token proportion) $k\%$ to separate between informative and uninformative tokens. Denote by $\hat{y}_{i,j}$ the token label after cleaning. We have

$$\hat{y}_{i,j} = \begin{cases} 1 & \text{if } \mathsf{Score}(x_{i,j}|\boldsymbol{x}_{i,:j};\theta,\theta') \text{ ranks top } k\%, \forall i,j; \\ 0 & \text{otherwise.} \end{cases} \tag{4}$$

### 4.3. Selection of $\theta$ and $\theta'$

We discuss two feasible strategies for selecting $\theta$ and $\theta'$.

### 4.3.1. Fixed-Model Cleaning

Following the analyses in Section 4.2.1, we can assume the access to a model $\theta'$ that outperforms $\theta$. For example, a moderately performing Llama model can be considered as $\theta$, while a well-performing Llama model can be considered as $\theta'$ (Mindermann et al., 2022; Lin et al., 2024). Specifically, given the warm-up model $\theta'$ and the base model $\theta$, we compute the token scores for the entire dataset $\widetilde{D}$ according to Eq. (3) and use a fixed threshold $k_{\text{fixed}}$ to assign token labels $\tilde{y}_{i,j}$ according to Eq. (4). Note that token cleaning is performed globally, meaning that some samples may be entirely removed if they contain no positive tokens. This differs from Lin et al. (2024), where each sample retains a fixed proportion of positive tokens. The benefits and limitations of this strategy will be discussed theoretically in Section 5.2.

### 4.3.2. Self-Evolving Cleaning

Inspired by the success of semi-supervised learning (SSL), we propose to do token cleaning iteratively. Specifically, in the $t$-th iteration, we fix the base model $\theta$ and adopt $\theta' = \theta_t$, then fine-tune $\theta_t$ with the selected tokens after cleaning. See Algorithm 1 for more details.

**Algorithm Details** The overall procedure is outlined in Algorithm 1. First, we evenly partition the dataset $\widetilde{D}$ into a series of subsets, denoted as $\{\widetilde{D}_0, \widetilde{D}_1, \ldots, \widetilde{D}_T\}$. Next, the base model $\theta$ is fine-tuned on the initial subset $\widetilde{D}_0$ to produce a warm-up model $\theta_0$, which serves as the initial reference model. Rather than relying solely on $\theta_0$ as a fixed reference model, a self-evolving mechanism is introduced in Lines 4-8. Specifically, for each subsequent subset $\widetilde{D}_t$, we keep the base model fixed and utilize the latest updated model as the reference model, i.e., $\theta = \theta_0$ and $\theta' = \theta_t$, to compute token scores. By applying a threshold $k_{\text{self-evol}}$ to these scores, we obtain the cleaned labels $\hat{y}_{i,j}$. The updated model $\theta_t$ is then fine-tuned on the cleaned subset $\widehat{D}_t$, producing the reference model for the next iteration. This process continues iteratively, and the final reference model is used as the output of the algorithm.

**Algorithm 1** Token Cleaning Pipeline

1: **Input:** Entire dataset $\widetilde{D}$, base model $\theta_0$, threshold $k_{\text{self-evol}}$.
2: Split dataset $\widetilde{D}$ into a series of subset $\{\widetilde{D}_0, \cdots, \widetilde{D}_T\}$. Denote their indices by $\{S_0, \cdots, S_T\}$.
3: Warmup Model $\theta_1$: Finetune from base model $\theta_0$ on $\widetilde{D}_0$ subset with all tokens.
4: **for** $t$ **in** $\{1, 2, \cdots, T\}$ **do**
5:     Compute scores for subset $\widetilde{D}_t$'s tokens via $\mathsf{Score}(x_{i,j}|\boldsymbol{x}_{i,:j}; \theta_{t-1}, \theta_t), \forall x_{i,j} \in \widetilde{D}_t$.
6:     Assign token labels $\hat{y}_{i,j}$ with a threshold $k_{\text{self-evol}}$.
7:     Obtain $\theta_{t+1}$ by finetuning $\theta_t$ on cleaned subset $\widehat{D}_t = \{(x_{i,j}, \boldsymbol{x}_{i,:j}, \hat{y}_{i,j}), \forall(i,j) \in S_t\}$.
8: **end for**
9: **Output:** $\theta_{T+1}$

## 5. Theoretical Analyses

Let $\mathbb{1}\{\cdot\}$ be the indicator function taking value 1 when the specified condition is satisfied and 0 otherwise. Define the *0-1 loss* as $\mathbb{1}(\theta(\boldsymbol{X}_{\text{prev}}), X_{\text{next}}) := \mathbb{1}\{\theta(\boldsymbol{X}_{\text{prev}}) \neq X_{\text{next}}\}$, where $X_{\text{next}}$ is the random variable for the next token, $\boldsymbol{X}_{\text{prev}}$ is the random variable for tokens before the next token, and $\theta(\boldsymbol{X}_{\text{prev}})$ stands for the prediction of next token for model $\theta$ given $\boldsymbol{X}_{\text{prev}}$ as input. Without loss of generality, we consider the ideal case where all the training instances for next-token prediction are *i.i.d.* and minimize 0-1 loss in the following analyses. The loss can be generalized to bounded loss $\ell(\cdot)$ and finite function space $\mathcal{F}$ following the generalization bounds that can be introduced using Rademacher complexity (Bartlett & Mendelson, 2002).

### 5.1. Exceed the Performance of Full Tokens

Denote by $\widetilde{D} := \{(x_{i,j}, \boldsymbol{x}_{i,:j}, \tilde{y}_{i,j}), \forall i, j\}$ the full-token dataset. By minimizing the noisy loss

$$\widehat{\mathcal{L}}_{\widetilde{D}}(\theta) = \frac{1}{N} \sum_{i=1}^{N} \frac{1}{\sum_{j=1}^{L_i} \tilde{y}_{i,j}} \sum_{j=1}^{L_i} \tilde{y}_{i,j} \mathbb{1}(\theta(\boldsymbol{x}_{i,:j}), x_{i,j})$$

we can get model $\hat{\theta}_{\widetilde{D}} := \arg\min_{\theta} \widehat{\mathcal{L}}_{\widetilde{D}}(\theta)$. Denote by $\widetilde{Y}, Y$ the random variables for $\tilde{y}_{i,j}$ and the corresponding ground-truth label $y_{i,j}$. The expected loss of training with full tokens can be denoted by

$$\mathcal{L}_{\widetilde{\mathcal{D}}}(\theta) = \mathbb{E}\left[\widetilde{Y} \cdot \mathbb{1}(\theta(\boldsymbol{X}_{\text{prev}}), X_{\text{next}})\right],$$

where $\widetilde{\mathcal{D}}$ is the distribution of $\widetilde{D}$. Denote by

$$\eta(\widetilde{D}) := \mathbb{P}(\widetilde{Y} \neq Y)$$

the noise rate of full tokens. Theorem 5.1 shows the error upper bound of learning with full tokens. See Appendix C for the proof.

**Theorem 5.1** (Error of learning with full tokens). *With probability at least $1 - \delta$, the generalization error of learning with full tokens is upper-bounded by*

$$\mathcal{L}_{\mathcal{D}}(\hat{\theta}_{\widetilde{D}}) \leq \underbrace{\eta(\widetilde{D})}_{\text{Data quality}} + \underbrace{\sqrt{\frac{2\log(4/\delta)}{M}}}_{\text{Data quantity}}, \quad (5)$$

*where $M := \sum_{i=1}^{N} L_i$ denotes the number of tokens.*

Theorem 5.1 shows that the error of learning with full tokens depends on two factors:

- **Data Quality**: $\eta(\widetilde{D})$ denotes the noise rates of learning with full tokens, where a higher noise rate leads to a larger error, i.e., a worse performance. The negative impact of wrong token labels may not be canceled by increasing the number of tokens.
- **Data Quantity**: When the number of tokens $M$ increases, the generalization could be smaller, showing that the token cleaning result cannot merely return a small set of high-quality tokens, i.e., the precision and recall of the token cleaning algorithm are both important.

Denote by $\widehat{Y}$ the random variable of token labels after cleaning. We show the superiority of token cleaning compared to the full-token case in the following corollary.

**Corollary 5.2.** *With probability as least $1 - 2\delta$, token cleaning performs better than full-tokens in terms of the error upper bound when*

$$\eta(\widetilde{D}) - \hat{\eta} \geq \sqrt{2\log(4/\delta)} \cdot \sqrt{\frac{1}{M}} \cdot \left(\sqrt{\frac{1}{\hat{r}}} - 1\right), \quad (6)$$

*where $\hat{\eta} := \mathbb{P}(\widehat{Y} \neq Y)$ denotes the noise rates of cleaned labels and $\hat{r} := \mathbb{P}(\widehat{Y} = 1)$ denotes the ratio of positive tokens after token cleaning.*

Corollary 5.2 shows token cleaning is preferred when the positive impact of reducing noise rate outweighs the negative impact of reducing the number of feasible tokens. For example, when $M$ is larger (a larger dataset), the inequality in Corollary 5.2 is more likely to hold since the right-hand side is smaller.

### 5.2. Fixed-Model Cleaning: Stable But Limited Improvement

We now analyze the benefits and limitations based on Theorem 5.1 and Corollary 5.2. By selecting an appropriate model $\theta'$, we can take a one-shot token cleaning on all the tokens in the candidate data pool. In this case:

- **Data Quality**: The noise rate of cleaned tokens is fixed, i.e., the *data quality* term in Eq. (5) is fixed. By carefully selecting the threshold $k_{\text{fixed}}$, there exists a token cleaning result whose noise rate $\hat{\eta}$ is less than $\eta(\widetilde{D})$.

- **Data Quantity**: With more tokens being cleaned, $M$ is increasing. Then the total generalization error can be consistently reduced.

Therefore, under this strategy, as long as the reference model $\theta'$ is sufficiently good to reduce the noise rate from $\eta(\widetilde{D})$ to a lower rate $\hat{\eta}$, the model's performance can be improved by fine-tuning with additional *i.i.d.* cleaned tokens, demonstrating the advantage on stability. However, even as $M \to \infty$, the total error does not go to zero due to imperfect data quality, showing the limitations on final performance.

### 5.3. Self-Evolving Cleaning: Potential Matthew Effect

For ease of presentation, we divide the data into three groups according to their task difficulty and number of *i.i.d.* clean tokens in the training dataset:

- $G_1$ (Rich Group): Characterized by lower noise rates after token cleaning and a higher proportion of effective tokens. This group typically experiences significant performance gains during warmup (Line 3, Algorithm 1) and has a great number of relevant tokens.
- $G_2$ (Poor Group): Marked by higher noise rates after token cleaning and fewer effective tokens. This group often exhibits limited or even degraded performance during warmup and has a scarce number of relevant tokens.
- $G_3$ (Intermediate Group): Falling between the rich and poor groups in terms of data quality and quantity. While it generally sees reasonable performance improvement during warmup, its convergence tends to be unstable due to a limited number of effective tokens.

Note that the definition of groups only applies to the theoretical analyses, which does not mean we need to explicitly know the group attribute of each data. In fact, it is challenging to know this information. Theoretically, there are three observations during SFT.

- *Observation 1: The rich get richer ($G_1$).* When $\hat{\eta} < \eta(\widetilde{D})$ and $\hat{r} \cdot M$ is sufficiently large, according to Corollary 5.2, fine-tuning on $\widehat{D}_1$ benefits from lower token noise rates and a higher number of effective tokens, thereby reducing the error upper bound and resulting in a better model $\theta_2$. With a better reference model and a similar number of effective tokens, the model in the next iteration can be further improved, i.e., the rich get richer.
- *Observation 2: The poor get poorer ($G_2$).* When the model $\theta_1$ underperforms compared to $\theta_0$, $\eta_1$ increases significantly and may even exceed $0.5$. According to Corollary 5.2, continued fine-tuning on tokens with such high noise rates can further degrade performance, exemplifying a "the poor get poorer" effect.
- *Observation 3: Unstable convergence ($G_3$).* If $\hat{\eta}$ is comparable to $\eta(\widetilde{D})$ and $\hat{r} \cdot M$ is only marginally sufficient, fine-tuning on $\widehat{D}_3$ may yield moderate improvements but

suffers from instability due to the limited number of effective tokens. According to Corollary 5.2, the model's error upper bound may not decrease consistently, leading to unstable convergence.

From the above analyses, we know that self-evolving models are more adaptive and aggressive compared to fixed models. The theoretical insights further highlight strategies for achieving better performance.

## 6. Experiments

### 6.1. Experiments Setup

**Data Pool** We utilize a high-quality data pool with 50k sample size from five popular SFT datasets (300k in total): Flan_v2 (Longpre et al., 2023), Open Assistant 1 (Köpf et al., 2024), Stanford Alpaca (Taori et al., 2023), Dolly (Databricks, 2023), and WizardLM (Xu et al., 2023). The data pool is constructed based on a new powerful data curation pipeline proposed by (Pang et al., 2024b), which involves selecting data samples using quality rating scores generated by LLMs. More dataset statistical information including token length can be found in Appendix D.1. For the self-evolving cleaning strategy, we heuristically divide the data pool into five equally sized subsets (10k samples).

**Base Models** In this paper, we select three popular open-source LLMs as our base models, including LLaMA-3.2-3B, LLaMA-3.1-8B (Dubey et al., 2024) and Mistral-7B-v0.3 (Jiang et al., 2023). These base models will be fine-tuned using samples from our data pool.

**Baselines** There are several baselines for performance comparisons: 1) BASE denotes the used base model; 2) DS$^2$ (Pang et al., 2024b) fine-tunes base model on 10k selected high-quality samples (with full tokens) from the entire data pool (50k); 3) FULL TOKENS utilizes all tokens to fine-tune the base model; 4) UNIFORM RANDOM randomly selects $k\%$ tokens from the 50k data pool without replacement; 5) RHO (Mindermann et al., 2022; Lin et al., 2024) directly computes the excess loss for all tokens between the base and reference model and then selects top-$k\%$ tokens. Recall that $k$ is the pre-defined threshold for token cleaning.

**Warmup** We warmup by fine-tuning the base model on subset $\widetilde{D}_0$ with full tokens, and make it the (initial) reference model for RHO, our FIXED-MODEL CLEANING, and SELF-EVOLVING CLEANING. The warmup model is equivalent to the DS$^2$ baseline (Pang et al., 2024b).

**Evaluation** To comprehensively evaluate the efficacy of token cleaning methods, we adopt seven OpenLLM Leaderboard tasks, including MMLU (Hendrycks et al., 2020), TruthfulQA (Lin et al., 2021), TydiQA (Clark et al., 2020), HellaSwag (Zellers et al., 2019), ARC-Challenge (Clark et al., 2018), BoolQ (Clark et al., 2019) and LogiQA. These

Table 1: Performance (original score ×100) comparison of different baselines on various benchmarks. We highlight the best result in **boldface** and the second-best with underline. By default, the selected token proportion (i.e., threshold) is 0.6.

| Model | TruthfulQA | TydiQA | LoqiQA | MMLU | HellaSwag | ARC-C | BoolQ | AVG |
|---|---|---|---|---|---|---|---|---|
| Base model: LLaMA-3.2-3B | | | | | | | | |
| BASE | 39.39 | 21.10 | 22.17 | 56.29 | 55.24 | 42.20 | 72.95 | 44.19 |
| DS$^2$ (10K) | 43.35 | 41.20 | 24.96 | 56.93 | 55.64 | 44.62 | 74.80 | 48.79 |
| FULL TOKENS (50K) | 43.32 | 49.60 | 24.34 | 56.87 | 55.57 | 44.44 | 74.98 | 49.87 |
| UNIFORM RANDOM (50K×0.6) | 43.79 | 47.00 | 23.41 | 56.96 | 55.37 | 44.44 | 75.05 | 49.43 |
| RHO | 45.57 | 53.60 | 26.05 | **57.10** | 55.16 | 45.39 | 77.36 | 51.46 |
| FIXED-MODEL CLEANING | 48.96 | 52.60 | 25.89 | 57.09 | **56.43** | 45.39 | **77.52** | 51.98 |
| SELF-EVOLVING CLEANING | **51.07** | **56.38** | **28.22** | 56.18 | 55.81 | **45.99** | 77.33 | **53.00** |
| Base model: LLaMA-3.1-8B | | | | | | | | |
| BASE | 45.10 | 22.80 | 26.51 | 65.29 | 59.92 | 50.82 | 82.18 | 50.37 |
| DS$^2$ (10K) | 49.57 | 45.80 | 27.44 | 65.77 | 60.37 | 53.49 | **83.26** | 55.10 |
| FULL TOKENS (50K) | 47.51 | 58.10 | 28.53 | 65.78 | 60.42 | 54.01 | 82.49 | 56.70 |
| UNIFORM RANDOM (50K×0.6) | 48.68 | 56.60 | 27.29 | **65.81** | 60.40 | 54.09 | 83.11 | 56.57 |
| RHO | 54.63 | 61.90 | **28.99** | 65.74 | 62.14 | 54.78 | 81.66 | 58.55 |
| FIXED-MODEL CLEANING | 56.02 | 62.38 | 28.22 | 65.71 | 61.92 | **55.12** | 82.67 | 58.90 |
| SELF-EVOLVING CLEANING | **59.58** | **63.58** | 26.05 | 65.07 | **62.67** | 54.87 | 82.49 | **59.20** |
| Base model: Mistral-7B-v0.3 | | | | | | | | |
| BASE | 42.56 | 54.70 | 25.74 | 62.41 | 60.77 | 48.92 | 82.30 | 52.88 |
| DS$^2$ (10K) | 44.24 | 55.70 | 25.27 | **62.50** | 61.10 | 50.39 | 83.45 | 53.85 |
| FULL TOKENS (50K) | 43.67 | 55.60 | 25.27 | 62.41 | 61.14 | 50.56 | **83.85** | 54.12 |
| UNIFORM RANDOM (50K×0.6) | 43.82 | 55.70 | 24.81 | 62.47 | 61.20 | 50.04 | 83.76 | 53.64 |
| RHO | 43.92 | 54.50 | 25.43 | 62.12 | 61.35 | 51.08 | 83.29 | 53.60 |
| FIXED-MODEL CLEANING | 44.52 | **59.03** | 26.05 | 61.45 | **61.47** | **51.68** | 82.03 | **55.20** |
| SELF-EVOLVING CLEANING | **45.41** | 56.17 | **27.44** | 62.30 | 61.40 | 50.65 | 81.28 | 55.00 |

datasets are sufficiently diverse to thoroughly assess the fine-tuned model across various aspects, including factual accuracy, reasoning, and multilingual capability. The task performances are evaluated on the lm-eval-hareness[1] repository. More evaluation and training details can be found in Appendix D.2.

## 6.2. Main Empirical Results

As shown in Table 1, our proposed strategies consistently outperform baselines across three base models on seven evaluation benchmarks. Notably, compared to using full tokens, our self-evolving cleaning has achieved the average performance improvement of **6.3%** on the 3B model and **2.0%/4.4%** on the 7B/8B models.

**Local Ranking vs. Global Ranking**   Compared to RHO (Lin et al., 2024), which ranks token scores locally within individual samples and removes the same proportion of tokens per sample, our fixed-model cleaning method globally ranks token scores across the entire dataset. As shown in Table 1,

the local-ranking method (RHO) yields lower average performance than the global-ranking method (FIXED-MODEL CLEANING), e.g., 51.46 vs. 51.98 for LLaMA-3.2-3B and 53.60 vs. 55.20 for Mistral-7B-v0.3, demonstrating that global ranking leads to more stable performance improvements. One possible explanation is that local ranking is constrained by the quality of individual samples. For example, in SFT, a low-quality sample may not contain any useful tokens, while almost all tokens in a high-quality sample may be useful. Since local ranking removes the same proportion of tokens from both samples, it inevitably retains uninformative tokens from low-quality samples while discarding informative ones from high-quality samples. This limitation can be mitigated through global token ranking, as employed in our fixed-model cleaning approach.

**Self-Evolving Cleaning Follows the Matthew Effect**   Table 2 presents the model's performance across different training iterations (checkpoints), illustrating three phenomena arising from the Matthew effect, as discussed in Section 5.3. Specifically, performance on TruthfulQA, TydiQA, and LogiQA steadily improves over iterations, representing *Observation 1: the rich get richer*. In contrast, MMLU

[1]https://github.com/EleutherAI/lm-evaluation-harness

Table 2: Performance results of self-evolving cleaning strategy over iterations (checkpoints) on seven benchmarks. Base model: `LLaMA-3.2-3B`. These performance results align with three observations arising from the Matthew effect.

| Model | TruthfulQA | TydiQA | LoqiQA | MMLU | HellaSwag | ARC-C | BoolQ |
|---|---|---|---|---|---|---|---|
| REFERENCE-1 | 45.46 | 50.05 | 27.44 | 57.31 | 56.10 | 45.91 | 76.87 |
| REFERENCE-2 | 46.67 | 53.18 | 27.44 | 56.89 | 56.25 | 46.51 | 77.15 |
| REFERENCE-3 | 48.91 | 54.36 | 28.22 | 56.43 | 56.13 | 46.43 | 77.36 |
| REFERENCE-4 | 51.07 | 56.38 | 28.22 | 56.18 | 55.81 | 45.99 | 77.33 |

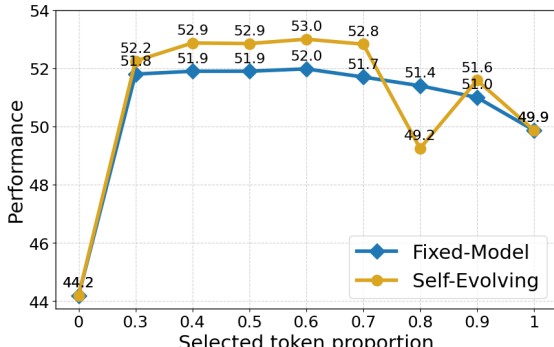

Figure 2: Average performance results of two cleaning pipelines under different token proportions. Base model: LLaMA-3.2-3B.

which focuses on factual knowledge, exhibits a slight performance decline, illustrating *Observation 2: the poor get poorer*. Meanwhile, for the remaining tasks (HellaSwag, ARC-C, and BoolQ), performance improvements are observed but exhibit fluctuations, aligning with *Observation 3: unstable convergence*.

### 6.3. Ablation Study

**Impact of Selected Token Proportion** Here, we investigate the impact of selected token proportion for our pipeline using a series of token proportion values including $\{0.3, 0.4, \cdots, 0.9\}$. As shown in Figure 2, the best results are achieved when the selected token proportion is approximately 50% to 70%. Beyond this range, the overall performance declines, which may be attributed to uninformative tokens. One valuable empirical finding is that *the performance gains in SFT tasks largely rely on a small number of highly informative and clean tokens*. This observation supports the prevailing view that data quality is more crucial than mere volume. Full results can be referred to in Appendix E (Table 7).

**Impact of Reference Model** To assess the impact of the reference model on performance, we run RHO and our fixed-model cleaning approach using LLaMA-3.1-8B-Instruct as the new reference model and compare the results with our previous reference model (DS$^2$ as the warmup). As shown

Table 3: Performance comparison with a new reference model: LLaMA-3.1-8B-Instruct. Blue: A better reference model brings a higher performance improvement. Red: The counterpart.

| Models | MMLU | BoolQ | TydiQA | ARC-C |
|---|---|---|---|---|
| REFERENCE: LLAMA-8B-INST | 68.18 | 84.03 | 21.63 | 51.77 |
| RHO | 57.04 | 75.94 | 39.37 | 46.08 |
| FIXED-MODEL CLEANING | 56.96 | 76.37 | 39.17 | 46.08 |
| REFERENCE: WARMUP | 56.93 | 74.80 | 41.20 | 44.62 |
| RHO | 57.10 | 77.36 | 53.60 | 45.39 |
| FIXED-MODEL CLEANING | 57.09 | 77.52 | 52.60 | 45.39 |

in Table 3, a more powerful reference model generally leads to greater performance improvements in datasets such as TydiQA and ARC-C. However, some counterintuitive results emerge: in MMLU and BoolQ, despite the 8B reference model significantly outperforming the warmup model, it fails to yield further improvements through token cleaning. A possible explanation for this phenomenon is the distribution shift. If we divide the evaluation task distribution into two parts: an in-distribution segment aligned with our data pool and an out-of-distribution segment, LLaMA-3.1-8B-Instruct, while achieving high overall performance, may not necessarily surpass the warmup model in the in-distribution subset. Investigating this hypothesis and validating this assumption are promising directions for future research. More detailed results can be found in Appendix E.

### 7. Conclusion

This work has demonstrated the effectiveness and importance of token cleaning, introducing a generic token cleaning pipeline that removes uninformative tokens while preserving task-relevant information. Our theoretical analysis has revealed the strengths and limitations of two scoring strategies: *fixed-model cleaning*, which provides stability but limited improvements, and *self-evolving cleaning*, which has shown the potential for greater performance gains but requires more careful implementation. Empirically, we have found that filtering out approximately 30%–40% of tokens consistently enhances performance across both strategies, achieving an average 6.3% improvement over the full-token baseline at the 3B model scale.

## Acknowledgment

Pang and Qian were partially supported by NSF Grants 2322919, 2420632, 2426031, and 2426940.

## Impact Statement

This paper presents work whose goal is to advance the field of Machine Learning. There are many potential societal consequences of our work, none of which we feel must be specifically highlighted here.

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

# Appendix

## Organization of the Appendix

The Appendix is organized as follows.

- Section A discusses the potential limitations of this work.
- Section B discusses the additional computational cost of our proposed piplines.
- Section C provides a full proof for Theorems shown in Section 5.
- Section D illustrates the experimental details including data pool, evaluation benchmarks, and training details.
- Section E demonstrates detailed omitted performance results and the visualization of token influence scores.
- Section F provides several samples with tokens selected by the self-evolving cleaning pipeline.

## A. Limitations

While the proposed token cleaning pipelines demonstrate competitive performance compared to other baselines, we acknowledge that there are still potential limitations:

- **Base Model Scale** Our experiments are primarily conducted on a base model with a 3B-8B scale. It remains uncertain how well the pipeline would perform on larger-scale base models.

- **Data Pool Scale** Due to cost considerations, our data pool is limited to 50k samples. The performance of the proposed pipeline on a larger-scale data pool remains uncertain.

## B. Computational Time and Cost Analysis

**Computational Cost**    The computational costs associated with the token cleaning pipelines primarily consist of the training costs, akin to those of standard SFT, along with two types of additional inference costs. These additional costs stem from one base model and another reference model, both of which are used to calculate token-level influence scores. Compared to the Rho baseline (Lin et al., 2024), our two proposed piplines do not incur any additional inference costs since the total data size for inference remains unchanged. For the Naive Fixed-Model Cleaning, we perform a one-shot inference on all samples simultaneously, mirroring the process used in Rho. For the Self-Evolving Cleaning pipeline, we simply segment the data pool into several partitions for independent inference using different reference models, i.e., 50k samples $\rightarrow \{10\text{k}, 10\text{k}, \cdots, 10\text{k}\}$ samples. Consequently, the inference cost for the Self-Evolving Cleaning pipeline is also equivalent to that of Rho baseline.

**Computational Time**    It's important to note that compared to standard SFT, token cleaning pipelines does not achieve computational speedup because the next-token prediction $x_{ij}$ still relies on the context formed by previous tokens $\mathbf{x}_{i,:j}$. The current implementation simply masks out uninformative tokens to ignore their token loss, which is simple and compatible with all training paradigms. Existing token-level approaches, including RHO and our approaches, mainly focus on performance efficiency rather than data (token) training efficiency. In practice, the GPU memory usage of these methods remains the same as when using full tokens. Therefore, investigating and enhancing token training efficiency represents a promising direction for future research, such as exploring ways to skip uninformative token memory occupation altogether.

## C. Proof for Theorem 5.1

We first reproduce the definitions as follows.

Denote by $\widetilde{D} := \{(x_{i,j}, \boldsymbol{x}_{i,:j}, \tilde{y}_{i,j}), \forall i, j\}$ the full-token dataset. By minimizing the noisy loss

$$\widehat{\mathcal{L}}_{\widetilde{D}}(\theta) = \frac{1}{\sum_{(i,j)\in S} \tilde{y}_{i,j}} \sum_{(i,j)\in S} \tilde{y}_{i,j} \mathbb{1}(\theta(\boldsymbol{x}_{i,:j}), x_{i,j}),$$

where $S := \{(i,j) | i \in [N], j \in [L_i]\}$, $[N] := \{1, 2, \cdots, N\}$. we can get model $\hat{\theta}_{\widetilde{D}} := \arg\min_\theta \widehat{\mathcal{L}}_{\widetilde{D}}(\theta)$. When train with

full tokens, we have $\tilde{y}_{i,j} = 1, \forall i, j$. The corresponding expected loss can be denoted by

$$\mathcal{L}_{\widetilde{\mathcal{D}}}(\theta) = \mathbb{E}\left[\widetilde{Y} \cdot \mathbb{1}(\theta(\boldsymbol{X}_{\text{prev}}), X_{\text{next}})\right] = \mathbb{E}\left[\mathbb{1}(\theta(\boldsymbol{X}_{\text{prev}}), X_{\text{next}})\right],$$

where $\widetilde{\mathcal{D}}$ is the distribution of $\widetilde{D}$. Denote by $\widetilde{Y}$ the random variable for noisy token label $\tilde{y}_{i,j}$, and $Y$ the random variable for the ground-truth token label $y_{i,j}$. Accordingly, with ground-truth token labels, the expected loss is

$$\mathcal{L}_{\mathcal{D}}(\theta) = \frac{1}{\mathbb{E}[Y]}\mathbb{E}\left[Y \cdot \mathbb{1}(\theta(\boldsymbol{X}_{\text{prev}}), X_{\text{next}})\right].$$

**Decomposition**  With the above definitions, the generalization error of model $\hat{\theta}_{\widetilde{D}}$ on the clean distribution could be decomposed as

$$\mathcal{L}_{\mathcal{D}}(\hat{\theta}_{\widetilde{D}}) = \underbrace{(\mathcal{L}_{\mathcal{D}}(\hat{\theta}_{\widetilde{D}}) - \mathcal{L}_{\widetilde{\mathcal{D}}}(\hat{\theta}_{\widetilde{D}}))}_{\text{Term-1}} + \underbrace{\mathcal{L}_{\widetilde{\mathcal{D}}}(\hat{\theta}_{\widetilde{D}})}_{\text{Term-2}},$$

where **Term-1** transforms the evaluation of $\hat{\theta}_{\widetilde{D}}$ from clean distribution $\mathcal{D}$ to the noisy distribution $\widetilde{\mathcal{D}}$. **Term-2** is the generalization error but the model is trained and evaluated on noisy distribution $\widetilde{\mathcal{D}}$. Both terms are analyzed as follows.

### C.1. Term-1 Upper Bound

For a certain model $\theta$, there always exist a random variable $\widetilde{X}_{\text{next}}$ such that when $Y = 1$, $X_{\text{next}} = \widetilde{X}_{\text{next}}$ and when $Y = 0$,

$$\mathbb{E}_{X|Y=1}\left[\mathbb{1}(\theta(\boldsymbol{X}_{\text{prev}}), X_{\text{next}})\right] = \mathbb{E}_{X|Y=0}\left[\mathbb{1}(\theta(\boldsymbol{X}_{\text{prev}}), \widetilde{X}_{\text{next}})\right].$$

Then we have

$$\begin{aligned}
&\mathbb{E}\left[\widetilde{Y} \cdot \mathbb{1}(\theta(\boldsymbol{X}_{\text{prev}}), \widetilde{X}_{\text{next}})\right] \\
=&\mathbb{P}(Y = 1) \cdot \mathbb{E}_{X|Y=1}\left[\mathbb{1}(\theta(\boldsymbol{X}_{\text{prev}}), \widetilde{X}_{\text{next}})\right] + \mathbb{P}(Y = 0) \cdot \mathbb{E}_{X|Y=0}\left[\mathbb{1}(\theta(\boldsymbol{X}_{\text{prev}}), \widetilde{X}_{\text{next}})\right] \\
=&\mathbb{P}(Y = 1) \cdot \mathbb{E}_{X|Y=1}\left[\mathbb{1}(\theta(\boldsymbol{X}_{\text{prev}}), X_{\text{next}})\right] + \mathbb{P}(Y = 0) \cdot \mathbb{E}_{X|Y=1}\left[\mathbb{1}(\theta(\boldsymbol{X}_{\text{prev}}), X_{\text{next}})\right] \\
=&\mathbb{E}_{X|Y=1}\left[\mathbb{1}(\theta(\boldsymbol{X}_{\text{prev}}), X_{\text{next}})\right] \\
=&\frac{1}{\mathbb{E}[Y]} \cdot \mathbb{E}\left[Y \cdot \mathbb{1}(\theta(\boldsymbol{X}_{\text{prev}}), X_{\text{next}})\right]
\end{aligned}$$

$$\mathcal{L}_{\mathcal{D}}(\theta) - \mathcal{L}_{\widetilde{\mathcal{D}}}(\theta)$$

$$= \int_X \left( \frac{Y}{\mathbb{E}[Y]} \cdot \mathbb{P}(\theta(\boldsymbol{X}_{\text{prev}}) \neq X_{\text{next}}|X) - \widetilde{Y} \cdot \mathbb{P}(\theta(\boldsymbol{X}_{\text{prev}}) \neq X_{\text{next}}|X) \right) \mathbb{P}(X) \, dX$$

$$= \int_X \left( \widetilde{Y} \cdot \mathbb{P}(\theta(\boldsymbol{X}_{\text{prev}}) \neq \widetilde{X}_{\text{next}}|X) - \widetilde{Y} \cdot \mathbb{P}(\theta(\boldsymbol{X}_{\text{prev}}) \neq X_{\text{next}}|X) \right) \mathbb{P}(X) \, dX$$

$$= \int_X \left( \mathbb{P}(\theta(\boldsymbol{X}_{\text{prev}}) \neq \widetilde{X}_{\text{next}}|X) - \mathbb{P}(\theta(\boldsymbol{X}_{\text{prev}}) \neq X_{\text{next}}|X) \right) \mathbb{P}(X) \, dX$$

$$= \int_X \left( \mathbb{P}(\theta(\boldsymbol{X}_{\text{prev}}) = X_{\text{next}}|X) - \mathbb{P}(\theta(\boldsymbol{X}_{\text{prev}}) = \widetilde{X}_{\text{next}}|X) \right) \mathbb{P}(X) \, dX$$

$$\leq \frac{1}{2} \int_X \left( \left| \mathbb{P}(\theta(\boldsymbol{X}_{\text{prev}}) = \widetilde{X}_{\text{next}}|X) - \mathbb{P}(\theta(\boldsymbol{X}_{\text{prev}}) = X_{\text{next}}|X) \right| \right.$$

$$\left. + \left| \mathbb{P}(\theta(\boldsymbol{X}_{\text{prev}}) \neq \widetilde{X}_{\text{next}}|X) - \mathbb{P}(\theta(\boldsymbol{X}_{\text{prev}}) \neq X_{\text{next}}|X) \right| \right) \mathbb{P}(X) \, dX$$

$$\overset{(a)}{=} \int_X \text{TD}(\widetilde{X}_{\text{next}}(\boldsymbol{X}_{\text{prev}}; \theta) || X_{\text{next}}(\boldsymbol{X}_{\text{prev}}; \theta)) \mathbb{P}(X) \, dX$$

$$\overset{(b)}{\leq} \frac{1}{2} \int_X \sum_{k \in [K]} \left| \mathbb{P}(\widetilde{X}_{\text{next}} = k|X) - \mathbb{P}(X_{\text{next}} = k|X) \right| \mathbb{P}(X) \, dX$$

$$= \mathbb{P}(Y = 1) \cdot \frac{1}{2} \int_X \sum_{k \in [K]} \left| \mathbb{P}(\widetilde{X}_{\text{next}} = k|X, Y = 1) - \mathbb{P}(X_{\text{next}} = k|X, Y = 1) \right| \mathbb{P}(X|Y = 1) \, dX$$

$$+ \mathbb{P}(Y = 0) \cdot \frac{1}{2} \int_X \sum_{k \in [K]} \left| \mathbb{P}(\widetilde{X}_{\text{next}} = k|X, Y = 0) - \mathbb{P}(X_{\text{next}} = k|X, Y = 0) \right| \mathbb{P}(X|Y = 0) \, dX$$

$$\leq \mathbb{P}(Y = 0)$$

$$= \mathbb{P}(Y \neq \widetilde{Y})$$

$$= \eta(\widetilde{D}).$$

where in **equality (a)**, given model $\theta$ and previous tokens, we can treat $\widetilde{X}_{\text{next}}(\boldsymbol{X}_{\text{prev}}; \theta)$ as a Bernoulli random variable such that

$$\mathbb{P}(\widetilde{X}_{\text{next}}(\boldsymbol{X}_{\text{prev}}; \theta) = +) = \mathbb{P}(\theta(\boldsymbol{X}_{\text{prev}}) = \widetilde{X}_{\text{next}}|X) \text{ and } \mathbb{P}(\widetilde{X}_{\text{next}}(\boldsymbol{X}_{\text{prev}}; \theta) = -) = \mathbb{P}(\theta(\boldsymbol{X}_{\text{prev}}) \neq \widetilde{X}_{\text{next}}|X).$$

Then according to the definition of total variation of two distributions, i.e.,

$$\text{TD}(P||Q) := \frac{1}{2} \int_u |\frac{dP}{du} - \frac{dQ}{du}| du,$$

we can summarize the integrand as the total variation between $\widetilde{X}_{\text{next}}(\boldsymbol{X}_{\text{prev}}; \theta)$ and $X_{\text{next}}(\boldsymbol{X}_{\text{prev}}; \theta)$.

**Inequality (b)** holds due to the data processing inequality since the probabilities $[\mathbb{P}(\widetilde{X}_{\text{next}} = \theta(\boldsymbol{X}_{\text{prev}})), \mathbb{P}(\widetilde{X}_{\text{next}} \neq \theta(\boldsymbol{X}_{\text{prev}}))]$ are generated by $[\mathbb{P}(\widetilde{X}_{\text{next}} = i), \forall i \in [K]]$, and the probabilities $[\mathbb{P}(X_{\text{next}} = \theta(\boldsymbol{X}_{\text{prev}})), \mathbb{P}(X_{\text{next}} \neq \theta(\boldsymbol{X}_{\text{prev}}))]$ are generated by $[\mathbb{P}(X_{\text{next}} = k), \forall k \in [K]]$.

The results also hold for an arbitrary $\widetilde{Y}$. To prove it, we need to split the loss into two part: 1) $\widetilde{Y} = 1$ and 2) $\widetilde{Y} = 0$. The first part follows the previous proof, yields an upper bound of $\mathbb{P}(\widetilde{Y} = 1, Y = 0)$. The second part leads to an upper bound of $\mathbb{P}(\widetilde{Y} = 0, Y = 1)$. By summing up two bounds, we still have the same result, i.e., an upper bound of $\mathbb{P}(Y \neq \widetilde{Y})$.

## C.2. Term-2 Upper Bound

Recall that

$$\widehat{\mathcal{L}}_{\widetilde{D}}(\theta) = \frac{1}{\sum_{(i,j)\in S} \tilde{y}_{i,j}} \sum_{(i,j)\in S} \tilde{y}_{i,j} \ell(x_{i,j}|\boldsymbol{x}_{i,:j};\theta).$$

Let $\hat{\theta}_{\widetilde{D}}$ denote the model trained by minimizing 0-1 loss with noisy dataset $\widetilde{D}$, i.e.,

$$\hat{\theta}_{\widetilde{D}} := \arg\min_{\theta} \; \widehat{\mathcal{L}}_{\widetilde{D}}(\theta).$$

Recall that the expected error of model $\theta$ on distribution $\widetilde{\mathcal{D}}$ is

$$\mathcal{L}_{\widetilde{\mathcal{D}}}(\theta) = \mathbb{E}\left[\widetilde{Y} \cdot \mathbb{1}(\theta(\boldsymbol{X}_{\text{prev}}), X_{\text{next}})\right].$$

The optimal classifier is denoted by

$$\theta_{\widetilde{\mathcal{D}}} := \arg\min_{\theta} \mathcal{L}_{\widetilde{\mathcal{D}}}(\theta).$$

Denote by $M := \sum_{i=1}^{N} L_i$ the number of toekns. With probability at least $1 - \delta$, we have:

$$
\begin{aligned}
&\mathcal{L}_{\widetilde{\mathcal{D}}}(\hat{\theta}_{\widetilde{D}}) - \mathcal{L}_{\widetilde{\mathcal{D}}}(\theta_{\widetilde{\mathcal{D}}}) \\
=&\widehat{\mathcal{L}}_{\widetilde{D}}(\hat{\theta}_{\widetilde{D}}) - \widehat{\mathcal{L}}_{\widetilde{D}}(\theta_{\widetilde{\mathcal{D}}}) + \left(\mathcal{L}_{\widetilde{\mathcal{D}}}(\hat{\theta}_{\widetilde{D}}) - \widehat{\mathcal{L}}_{\widetilde{D}}(\hat{\theta}_{\widetilde{D}})\right) + \left(\widehat{\mathcal{L}}_{\widetilde{D}}(\theta_{\widetilde{\mathcal{D}}}) - \mathcal{L}_{\widetilde{\mathcal{D}}}(\theta_{\widetilde{\mathcal{D}}})\right) \\
\overset{(a)}{\leq}&0 + |\widehat{\mathcal{L}}_{\widetilde{D}}(\hat{\theta}_{\widetilde{D}}) - \mathcal{L}_{\widetilde{\mathcal{D}}}(\hat{\theta}_{\widetilde{D}})| + |\widehat{\mathcal{L}}_{\widetilde{D}}(\theta_{\widetilde{\mathcal{D}}}) - \mathcal{L}_{\widetilde{\mathcal{D}}}(\theta_{\widetilde{\mathcal{D}}})| \\
\overset{(b)}{\leq}&\sqrt{\frac{2\log(4/\delta)}{M}},
\end{aligned}
$$

where inequality (a) holds since 1) $\widehat{\mathcal{L}}_{\widetilde{D}}(\hat{\theta}_{\widetilde{D}})$ achieves the minimum empirical risk according to its definition, thus $\widehat{\mathcal{L}}_{\widetilde{D}}(\hat{\theta}_{\widetilde{D}}) - \widehat{\mathcal{L}}_{\widetilde{D}}(\theta_{\widetilde{\mathcal{D}}}) \leq 0$; 2) each of the following two terms will be no greater than the corresponding gap $|\widehat{\mathcal{L}}_{\widetilde{D}}(\theta) - \mathcal{L}_{\widetilde{\mathcal{D}}}(\theta)|$. Specifically, inequality (b) holds due to the Hoeffding's inequality, i.e., given any classifier $\hat{\theta}_{\widetilde{D}}$, and $\theta_{\widetilde{\mathcal{D}}}$, with probability at least $1 - \delta/2$, we have the following bounds independently:

$$|\widehat{\mathcal{L}}_{\widetilde{D}}(\hat{\theta}_{\widetilde{D}}) - \mathcal{L}_{\widetilde{\mathcal{D}}}(\hat{\theta}_{\widetilde{D}})| \leq \sqrt{\frac{\log(4/\delta)}{2M}}, \quad |\widehat{\mathcal{L}}_{\widetilde{D}}(\theta_{\widetilde{\mathcal{D}}}) - \mathcal{L}_{\widetilde{\mathcal{D}}}(\theta_{\widetilde{\mathcal{D}}})| \leq \sqrt{\frac{\log(4/\delta)}{2M}}.$$

By the union bound, we have inequality (b) with probability at least $1 - \delta$.

Supposing a unique next token for given previous tokens in distribution $\widetilde{\mathcal{D}}$, we have $\mathcal{L}_{\widetilde{\mathcal{D}}}(\theta_{\widetilde{\mathcal{D}}}) = 0$.

# D. Experimental Details

## D.1. Data Pool

The data pool used in this work is primarily composed of five widely used SFT datasets, derived either from human annotations or generated by advanced LLMs. Further details on these datasets can be found in Table 4. Notably, these datasets differ in format, quality, prompt length, and target tasks, highlighting the diversity of our foundational data pool. For consistency, we adopt the "Tulu" template format proposed by Wang et al. (2023) to standardize these datasets. The "Tulu" template includes two primary tags, `<|User|>` and `<|Assistant|>`, which denote the roles of the user and the assistant, respectively.

## D.2. Evaluation Benchmarks

For MMLU, TruthfulQA, LogiQA, ARC-C, BoolQ, and HellaSwag datasets, we directly use accuracy as the performance metric. In particular, for the TruthfulQA dataset (a multiple-choice question benchmark), we utilize the MC2 metric, which evaluates only the answer assigned the highest probability by the model (i.e., the model's most confident choice). For the TydiQA dataset, we consider using the 1-shot F1 score. One can double-check these performance metrics in the LM-EVALUATION-HARNESS repository. By default, we use all benchmark samples to conduct evaluation.

Table 4: Statistical summary of our 50k data pool. The average prompt length and the average response length are measured based on `LLaMA-3.2-3B`.

| Datasets | Data source | Data quality | # Data size | Prompt_Len | Completion_Len | Overall_Len |
|---|---|---|---|---|---|---|
| OPEN-ASSISTANT 1 | Human-generated | Both | 2418 | 45.4 | 364.1 | 409.5 |
| STANFORD ALPACA | Generated w/ Davinci-003 | Normal | 4598 | 22.5 | 84.8 | 107.2 |
| WIZARDLM | ChatGPT-generated | High | 34772 | 126.1 | 419.23 | 545.4 |
| DOLLY | Human-generated | Normal | 1567 | 119.1 | 133.7 | 252.9 |
| FLAN V2 | Human-generated | Normal | 6645 | 475.1 | 38.6 | 513.7 |

### D.3. Training Details

Following the experimental setup (Wang et al., 2023), we apply the LoRA technique (Hu et al., 2022) with a rank-size of 64 and a scaling factor of 16. The overall batch size is 48, with the learning rate at 1e-4 as well as 1 training epoch. By default, the maximum input length is 2048. All experiments are conducted on eight NVIDIA L40S GPUs. Fine-tuning 7B or 8B models on a dataset of 50k samples typically takes approximately 3 hours.

## E. Additional Experimental Results

### E.1. Empirical Results Still Follows the Observation of the Matthew Effort

Here, we also provide the empirical results of the self-evolving cleaning strategy on the other two base models. As shown in Table 5, the performance results over iterations also align with the theoretical observation, as discussed in Section 5.3.

Table 5: Performance results of self-evolving cleaning pipeline over iterations (checkpoints) on seven benchmarks. Base models: `LLaMA-3.1-8B` and `Mistral-7B-v0.3`. These performance results still align with three observations arising from the Matthew effect.

| Model | TruthfulQA | TydiQA | LoqiQA | MMLU | HellaSwag | ARC-C | BoolQ |
|---|---|---|---|---|---|---|---|
| **Base model: LLaMA-3.1-8B** | | | | | | | |
| REFERENCE-1 | 53.45 | 58.62 | 28.68 | 65.65 | 61.58 | 55.56 | 82.98 |
| REFERENCE-2 | 56.77 | 63.23 | 27.91 | 65.40 | 62.29 | 56.16 | 82.64 |
| REFERENCE-3 | 59.01 | 65.27 | 26.05 | 65.18 | 62.56 | 55.81 | 82.74 |
| REFERENCE-4 | 59.58 | 63.58 | 26.05 | 65.07 | 62.67 | 54.87 | 82.49 |
| **Base model: Mistral-7B-v0.3** | | | | | | | |
| REFERENCE-1 | 44.36 | 55.79 | 26.20 | 62.31 | 61.32 | 51.34 | 84.03 |
| REFERENCE-2 | 44.78 | 56.04 | 27.13 | 62.41 | 61.37 | 51.25 | 83.82 |
| REFERENCE-3 | 44.93 | 55.69 | 27.44 | 62.32 | 61.30 | 50.82 | 80.36 |
| REFERENCE-4 | 45.41 | 56.17 | 27.44 | 62.30 | 61.40 | 50.65 | 81.28 |

### E.2. Impact of Selected Token Proportion

This section provides more experimental results to explicitly demonstrate the impact of the selected token proportion across iterations. Here, Table 6 and Table 7 show the full performance results of the fixed-model cleaning and self-evolving cleaning pipeline, respectively.

### E.3. Impact of Reference Model

Here, we provide the full performance results under different reference models, as shown in Table 8. Here, we select `LLaMA-3.1-8B-Instruct`[2] as our reference model.

---
[2]https://huggingface.co/meta-llama/Llama-3.1-8B-Instruct.

Table 6: Performance comparison of self-evolving cleaning pipeline over different iterations on various benchmarks under different selected token proportions.

| Model | Token Prop | TruthfulQA | TydiQA | LoqiQA | MMLU | HellaSwag | ARC-C | BoolQ |
|---|---|---|---|---|---|---|---|---|
| **Base Model: LLaMA-3.2-3B** | | | | | | | | |
| REFERENCE-1 | | 45.68 | 49.36 | 27.60 | 57.23 | 56.06 | 45.56 | 77.08 |
| REFERENCE-2 | 0.3 | 46.84 | 50.10 | 28.22 | 56.53 | 56.01 | 47.11 | 77.18 |
| REFERENCE-3 | | 49.15 | 50.69 | 27.44 | 56.11 | 55.83 | 46.51 | 77.15 |
| REFERENCE-4 | | 51.57 | 52.96 | 26.51 | 55.42 | 55.55 | 46.51 | 77.24 |
| REFERENCE-1 | | 45.56 | 46.88 | 26.98 | 57.32 | 56.29 | 45.99 | 77.15 |
| REFERENCE-2 | 0.4 | 46.95 | 51.73 | 29.30 | 56.82 | 56.34 | 46.94 | 77.18 |
| REFERENCE-3 | | 48.72 | 54.56 | 28.84 | 56.52 | 56.05 | 46.68 | 77.52 |
| REFERENCE-4 | | 50.69 | 55.20 | 27.91 | 56.27 | 55.77 | 46.86 | 77.36 |
| REFERENCE-1 | | 45.47 | 47.49 | 27.13 | 57.18 | 56.22 | 45.82 | 76.99 |
| REFERENCE-2 | 0.5 | 46.93 | 53.02 | 27.44 | 57.04 | 56.45 | 46.68 | 77.30 |
| REFERENCE-3 | | 48.73 | 54.65 | 28.37 | 56.55 | 56.10 | 46.94 | 77.33 |
| REFERENCE-4 | | 50.42 | 55.19 | 28.37 | 56.25 | 55.83 | 46.77 | 77.12 |
| REFERENCE-1 | | 45.46 | 50.05 | 27.44 | 57.31 | 56.10 | 45.91 | 76.87 |
| REFERENCE-2 | 0.6 | 46.67 | 53.18 | 27.44 | 56.89 | 56.25 | 46.51 | 77.15 |
| REFERENCE-3 | | 48.91 | 54.36 | 28.22 | 56.43 | 56.13 | 46.43 | 77.36 |
| REFERENCE-4 | | 51.07 | 56.38 | 28.22 | 56.18 | 55.81 | 45.99 | 77.33 |
| REFERENCE-1 | | 45.34 | 50.45 | 27.60 | 57.42 | 56.09 | 45.65 | 76.90 |
| REFERENCE-2 | 0.7 | 46.82 | 52.02 | 26.98 | 56.86 | 56.34 | 46.43 | 77.30 |
| REFERENCE-3 | | 48.48 | 55.27 | 27.91 | 56.38 | 56.18 | 46.77 | 77.21 |
| REFERENCE-4 | | 50.66 | 55.47 | 27.91 | 56.10 | 55.93 | 46.43 | 77.36 |
| REFERENCE-1 | | 45.09 | 56.21 | 26.67 | 57.50 | 45.65 | 32.73 | 76.87 |
| REFERENCE-2 | 0.8 | 46.25 | 56.39 | 26.67 | 57.21 | 46.08 | 32.93 | 77.08 |
| REFERENCE-3 | | 47.30 | 56.34 | 26.67 | 57.05 | 45.99 | 33.33 | 77.33 |
| REFERENCE-4 | | 48.40 | 56.23 | 26.82 | 56.89 | 45.91 | 33.33 | 77.18 |
| REFERENCE-1 | | 44.75 | 46.88 | 26.20 | 57.52 | 56.10 | 45.39 | 76.68 |
| REFERENCE-2 | 0.9 | 45.06 | 50.47 | 26.20 | 57.27 | 56.42 | 46.08 | 76.96 |
| REFERENCE-3 | | 45.23 | 50.36 | 25.43 | 57.37 | 56.45 | 46.08 | 77.02 |
| REFERENCE-4 | | 46.36 | 51.68 | 26.98 | 57.25 | 56.34 | 45.65 | 76.90 |

### E.4. Token Influence Score Visualization

We visualize the token influence scores from our Self-Evolving Cleaning approach using scatter plots, as shown in Figure 3. As training progresses across iterations, we observe that token points gradually spread out from the diagonal line $y = x$, where most tokens were initially concentrated. This trend indicates that the influence scores become increasingly diverse. Figures 3a through 3b reveal similar token influence score patterns, as both base models originate from the LLaMA series, despite differing in scale. Specifically, more informative tokens (highlighted in red) shift into the upper-left region of the plot, where the reference model assigns low loss but the base model assigns higher loss. This pattern suggests that the Self-Evolving Cleaning strategy successfully amplifies the contrast in token-level influence scores across iterations, thereby enabling a more effective separation between informative and uninformative tokens.

## F. Illustrative Examples with Selected Tokens

Intuitively, those common tokens are less likely to be selected by our proposed strategies. Here, we provide several samples with tokens selected by the proposed self-evolving cleaning strategy. The selected (i.e., clean) tokens are highlighted in **blue**.

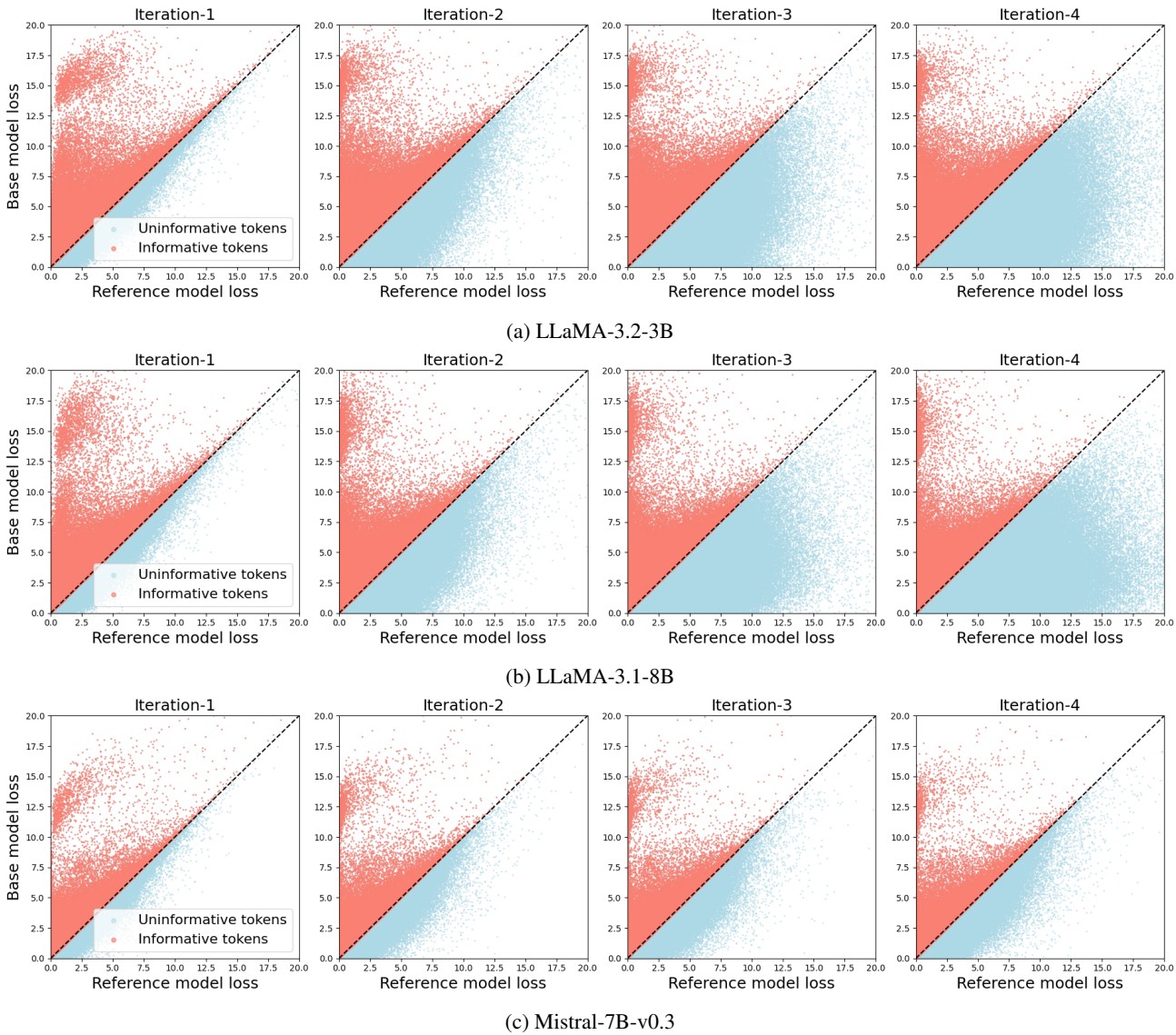

Figure 3: Visualization of token loss (i.e., influence scores) across training iterations of the Self-Evolving Cleaning pipeline. We plot the cases for three base models. The minimum influence score value of informative tokens is approximately in the range $[0.0003, 0.001]$. As the training progresses, token points diverge from the $y = x$ diagonal, indicating greater variability in influence scores. This allows Self-Evolving Cleaning to better differentiate between informative and uninformative tokens.

Table 7: Performance comparison of two token cleaning pipelines and RHO on various benchmarks under different selected token proportions. **Token Prop** denotes the selected token proportion.

| Model | Token Prop | TruthfulQA | TydiQA | LoqiQA | MMLU | HellaSwag | ARC-C | BoolQ | AVG |
|---|---|---|---|---|---|---|---|---|---|
| Base Model: LLaMA-3.2-3B | | | | | | | | | |
| FIXED MODEL CLEANNING | 0.3 | 48.46 | 51.08 | 26.36 | 56.93 | 56.22 | 46.25 | 77.36 | 51.80 |
| FIXED MODEL CLEANNING | 0.4 | 47.97 | 51.77 | 26.36 | 57.12 | 56.39 | 46.25 | 77.24 | 51.90 |
| FIXED MODEL CLEANNING | 0.5 | 47.65 | 52.61 | 26.51 | 57.14 | 56.45 | 45.99 | 77.15 | 51.90 |
| FIXED MODEL CLEANNING | 0.6 | 48.96 | 52.60 | 25.89 | 57.09 | 56.43 | 45.39 | 77.52 | 51.98 |
| FIXED MODEL CLEANNING | 0.7 | 47.53 | 51.96 | 25.74 | 57.06 | 56.55 | 45.99 | 77.15 | 51.70 |
| FIXED MODEL CLEANNING | 0.8 | 47.33 | 50.04 | 25.74 | 57.10 | 56.56 | 46.34 | 76.87 | 51.40 |
| FIXED MODEL CLEANNING | 0.9 | 46.26 | 49.75 | 24.81 | 57.18 | 56.39 | 46.34 | 76.03 | 51.00 |
| FIXED MODEL CLEANNING | 1.0 | 43.32 | 49.60 | 24.34 | 56.87 | 55.57 | 44.44 | 74.98 | 49.87 |
| SELF-EVOLVING CLEANING | 0.3 | 51.57 | 52.96 | 26.51 | 55.42 | 55.55 | 46.51 | 77.24 | 52.25 |
| SELF-EVOLVING CLEANING | 0.4 | 50.69 | 55.20 | 27.91 | 56.27 | 55.77 | 46.86 | 77.36 | 52.87 |
| SELF-EVOLVING CLEANING | 0.5 | 50.42 | 55.19 | 28.37 | 56.25 | 55.83 | 46.77 | 77.12 | 52.85 |
| SELF-EVOLVING CLEANING | 0.6 | 51.07 | 56.38 | 28.22 | 56.18 | 55.81 | 45.99 | 77.33 | 53.0 |
| SELF-EVOLVING CLEANING | 0.7 | 50.66 | 55.47 | 27.91 | 56.10 | 55.93 | 46.43 | 77.36 | 52.83 |
| SELF-EVOLVING CLEANING | 0.8 | 48.40 | 56.23 | 26.82 | 56.89 | 45.91 | 33.33 | 77.18 | 49.25 |
| SELF-EVOLVING CLEANING | 0.9 | 46.36 | 51.68 | 26.98 | 57.25 | 56.34 | 45.65 | 76.90 | 51.59 |
| SELF-EVOLVING CLEANING | 1.0 | 43.32 | 49.60 | 24.34 | 56.87 | 55.57 | 44.44 | 74.98 | 49.87 |
| RHO | 0.3 | 49.68 | 49.54 | 26.98 | 56.94 | 55.97 | 46.51 | 76.65 | 51.80 |
| RHO | 0.4 | 48.17 | 51.09 | 26.20 | 57.01 | 56.15 | 46.25 | 76.96 | 51.70 |
| RHO | 0.5 | 48.04 | 53.31 | 26.67 | 57.00 | 56.39 | 46.17 | 77.12 | 52.10 |
| RHO | 0.6 | 45.57 | 53.60 | 26.05 | 57.10 | 55.16 | 45.39 | 77.36 | 51.46 |
| RHO | 0.7 | 47.56 | 52.92 | 25.58 | 57.08 | 56.42 | 46.08 | 76.28 | 51.70 |
| RHO | 0.8 | 46.97 | 49.66 | 26.36 | 57.31 | 56.45 | 46.34 | 76.00 | 51.30 |
| RHO | 0.9 | 46.19 | 50.16 | 25.12 | 57.22 | 56.06 | 45.91 | 75.26 | 50.80 |
| RHO | 1.0 | 43.32 | 49.60 | 24.34 | 56.87 | 55.57 | 44.44 | 74.98 | 49.87 |

---

**Sample 1**

**User:** Write a response that appropriately completes the request.
### Instruction: Does Palmeiras have a Fifa club world cup?
### Response:

**Assistant:** Despite many claims that Palmeiras won a club world cup in 1951, the official list of the Fifa World Club winners starts in the year 2000 with Corinthians being the first champion of the tournament and Palmeiras doesn't have won any Fifa World Cup tournament since.

---

**Sample 2**

**User:** Seven Wonders of the Underwater World. Answer:

**Assistant:** Here's a list of the Seven Wonders of the Underwater World: - Palau- Belize Barrier Reef, Belize- Great Barrier Reef, Australia- Deep-sea hydrothermal vents (worldwide) - Galápagos Islands, Ecuador- Lake Baikal, Russia- Northern Red Sea, bordered by Saudi Arabia and Yemen on the eastern shore, and Egypt, Sudan, Eritrea, and Djibouti on the western shore

Table 8: Performance results under different reference models. Base model: LLaMA-3.1-3B. Additional reference model: LLaMA-3.1-8B-Instruct.

| Model | TruthfulQA | TydiQA | LoqiQA | MMLU | HellaSwag | ARC-C | BoolQ | AVG |
|---|---|---|---|---|---|---|---|---|
| BASE MODEL | 39.39 | 21.10 | 22.17 | 56.29 | 55.24 | 42.20 | 72.95 | 44.19 |
| LLaMA-8B-INST | 54.12 | 21.63 | 31.47 | 68.18 | 59.0 | 51.77 | 84.03 | 52.89 |
| RHO (LLAMA) | 44.06 | 39.37 | 25.58 | 57.04 | 55.35 | 46.08 | 75.94 | 49.1 |
| FIXED-MODEL cleaning (LLAMA) | 44.27 | 39.17 | 24.96 | 56.96 | 55.22 | 46.08 | 76.37 | 49.0 |
| WARMUP REFERENCE MODEL | 43.35 | 41.20 | 24.96 | 56.93 | 55.64 | 44.62 | 74.80 | 48.79 |
| RHO (WARMUP) | 45.57 | 53.60 | 26.05 | 57.10 | 55.16 | 45.39 | 77.36 | 51.46 |
| FIXED-MODEL cleaning (WARMUP) | 48.96 | 52.60 | 25.89 | 57.09 | 56.43 | 45.39 | 77.52 | 51.98 |

---

**Sample 3**

**User:** In a restaurant kitchen, a chef uses a food processor to prepare a sauce. The recipe calls for 2 cups of chopped tomatoes, 1 cup of diced onions, and 1/2 cup of minced garlic. The food processor has a maximum capacity of 4 cups. If the chef wants to make enough sauce for 6 servings, how many times must the ingredients be processed in the food processor? Hint: Assume that the food processor can process the entire recipe at once.

**Assistant:** The recipe requires a total of 2 + 1 + 0.5 = 3.5 cups of ingredients. To make enough sauce for 6 servings, the chef needs to prepare 6 * 3.5 = 21 cups of ingredients in total. Since the food processor can process up to 4 cups of ingredients at a time, the chef needs to process the ingredients 21 / 4 = 5.25 times. However, since the chef cannot process ingredients a fraction of a time, they will need to process the ingredients 6 times to make enough sauce for 6 servings.

---

**Sample 4**

**User:** What is the meaning of "real life"? How does one distinguish the real world from the imaginary? Can virtual reality experiences be considered real life? How do our senses interpret reality?

**Assistant:** The meaning of "real life" can be subjective and varies from person to person. Generally, it refers to experiences and interactions that occur in the physical world, as opposed to in one's imagination or virtual reality. Distinguishing the real world from the imaginary can be difficult, as imagination and perceptions can be influenced by personal beliefs, emotions, and past experiences. However, in general, the real world is considered to be the physical environment we inhabit and experience through our senses, whereas the imaginary world is one that exists solely in our minds and is not physically present. Virtual reality experiences can provide a sense of immersion that can feel very real, but they are still created and controlled by technology, and not experienced in the physical world. As such, they are not typically considered "real life." Our senses play a critical role in interpreting and experiencing reality. Our brains receive information from our eyes, ears, nose, skin, and other sensory receptors, and then process and interpret that information to create our experience of the world. However, our senses are not perfect and can be influenced by factors such as fatigue, illness, or external stimuli, which can alter our perceptions of reality.

## Sample 5

**User:** Develop a comprehensive metric that takes into consideration various factors such as user engagement, user feedback, user retention, user conversion rate, and user satisfaction to evaluate the effectiveness of a given phone application designed to connect people with mental health professionals. The metric should also take into account the different types of mental health conditions that the application caters to, such as depression, anxiety, bipolar disorder, and schizophrenia, and should provide separate scores for each condition. Additionally, the metric should consider the geographical location of the users and the availability of mental health professionals in their area. The evaluation should be done over a period of at least six months to account for any seasonal variations. Finally, the metric should be presented in a format that is easily understandable by both technical and non-technical stakeholders, such as a dashboard with charts and graphs, and should also include recommendations for improving the application based on the evaluation results.

**Assistant:** The following is a comprehensive metric that takes into consideration various factors to evaluate the effectiveness of a given phone application designed to connect people with mental health professionals: 1. User Engagement: This factor measures the level of user interaction with the application. The following metrics can be used to evaluate user engagement: - Number of downloads: The total number of downloads of the application over a period of six months. - Active Users: The number of users who actively use the application on a weekly or monthly basis. - Session Duration: The average time spent by users on the application during each session. - Frequency of Use: The number of times users access the application in a week or month. 2. User Feedback: This factor measures the level of satisfaction of users with the application. The following metrics can be used to evaluate user feedback: - App Rating: The average rating of the application on app stores such as Google Play Store and Apple App Store. - User Reviews: The number and content of user reviews on app stores and other forums. - Net Promoter Score (NPS): A metric that measures the likelihood of users to recommend the application to others. 3. User Retention: This factor measures the ability of the application to retain users over a period of time. The following metrics can be used to evaluate user retention: - Churn Rate: The percentage of users who stop using the application over a period of time. - Retention Rate: The percentage of users who continue to use the application over a period of time. 4. User Conversion Rate: This factor measures the ability of the application to convert users into paying customers. The following metrics can be used to evaluate user conversion rate: - Conversion Rate: The percentage of users who upgrade to a paid subscription or purchase additional services. - Average Revenue Per User (ARPU): The average amount of revenue generated per user. 5. User Satisfaction: This factor measures the level of satisfaction of users with the mental health professionals they connect with through the application. The following metrics can be used to evaluate user satisfaction: - Patient Satisfaction Score (PSS): A metric that measures the satisfaction of patients with the mental health professionals they connect with through the application. - Mental Health Professional Satisfaction Score (MHPSS): A metric that measures the satisfaction of mental health professionals with the application and the quality of patients they connect with. 6. Mental Health Condition: This factor takes into account the different types of mental health conditions that the application caters to, such as depression, anxiety, bipolar disorder, and schizophrenia. The following metrics can be used to evaluate the effectiveness of the application for each condition: - Condition-specific Engagement: The level of engagement of users with the application for each mental health condition. - Condition-specific Feedback: The level of satisfaction of users with the application for each mental health condition. - Condition-specific Retention: The ability of the application to retain users with each mental health condition over a period of time. - Condition-specific Conversion Rate: The ability of the application to convert users with each mental health condition into paying customers. - Condition-specific Satisfaction: The level of satisfaction of users with the mental health professionals they connect with through the application for each mental health condition. 7. Geographical Location: This factor takes into account the geographical location of the users and the availability of mental health professionals in their area. The following metrics can be used to evaluate the effectiveness of the application for each geographical location: - Location-specific Engagement: The level of engagement of users with the application for each geographical location. - Location-specific Feedback: The level of satisfaction of users with the application for each geographical location. - Location-specific Retention: The ability of the application to retain users in each geographical location over a period of time. - Location-specific Conversion Rate: The ability of the application to convert users in each geographical location into paying customers. - Location-specific Satisfaction: The level of satisfaction of users with the mental health professionals they connect with through the application in each geographical location. The evaluation should be done over a period of at least six months to account for any seasonal variations.

**Sample 6**

**User:** What's the furthest back we can prove human existence? Is there good reason to believe it goes further then this point?

**Assistant:** Human is an ever-evolving species. If we starts from Homo sapiens the species, there are traces of human in Africa since 300,000 years ago. For Homogenus, earliest fossil record discovered so far is dated around 2.8 million years ago. The above dates are based on fossil records. It is completely possible for new fossil emerges with an earlier date. It is also possible that carbon dating, technique used to determine fossil age, may not be accurate. For example if there are more carbon-14 than we thought there would be in the past, we would underestimate the age of existing fossils, pushing human history further back.

**User:** How reliable is carbon dating exactly?

**Assistant:** There is no exact answer to how reliable carbon dating is. It depends on many factors, such as the quality of the sample, the calibration of the equipment, and the assumptions made about the past. Generally speaking, carbon dating can be accurate up to about 50,000 years ago, but beyond that it becomes less reliable. The margin of error can range from a few years to several thousand years, depending on the situation.

