# OpenReview forum: "Token Cleaning: Fine-Grained Data Selection for LLM Supervised Fine-Tuning"
_ICML.cc/2025/Conference — ICML 2025 poster_

### Official Review · Reviewer_fypD · 2025-03-04

**Overall Recommendation:** 3

**Summary:**

This paper introduces two token cleaning methods for LLM fine-tuning, Fixed-model cleaning and Self-evolving cleaning. The token cleaning method ignores the loss from "unimportant" tokens, thereby improving task performance. The authors also provide theoretical analyses and extensive experiments.

**Claims And Evidence:**

Yes.

**Essential References Not Discussed:**

N/A

**Experimental Designs Or Analyses:**

Yes.

**Methods And Evaluation Criteria:**

Yes.

**Other Comments Or Suggestions:**

See Weaknesses

**Other Strengths And Weaknesses:**

## Strengths
- The paper is well written and easy to follow
- Theoretical analysis is interesting and adequately supports the findings
- Experiments are done thoroughly'

## Weaknesses
- My main concern is on the effectiveness of this approach. In Table 1, I am not sure if the author's approach is significantly better than previous work (e.g. RHO). I would like to see the average result from multiple runs and corresponding t-tests to see if the authors' approaches are consistently better.
- Another question is on the computational cost. It seems that multiple computations are needed in the token cleaning pipeline. How does the computation cost, e.g. in terms of latency, compare with vanilla SFT and other baselines?
- Although minor, I have some concerns regarding the technical contributions. The idea of removing unneeded tokens for fine-tuning is not new, and may somewhat limit the novelty of this work.

If all or part of my concerns are resolved, I am willing to re-evaluate my score.

**Questions For Authors:**

See Weaknesses

**Relation To Broader Scientific Literature:**

The work is very closely related to previous token selective fine-tuning methods. They differ from previous works by providing a semi-supervised style algorithm, self-evolving cleaning

**Theoretical Claims:**

Yes.

---

> ### Author Rebuttal · Authors · 2025-04-01
>
> We want to thank Reviewer fypD  for their positive feedback and comments. We will address individual comments below.
>
>
> > **Weakness 1: the effectiveness of this approach**
>
> **Response**:
> Thank you for your insightful suggestions！We present the average results from three independent trials (using three different random seeds) across two base model configurations. Clearly, our proposed methods still outperforms all baselines. Additionally, the corresponding t-tests results confirm that the performance improvements are significant.
>
> | Method(LLaMA-3.2-3B)| truthfulqa | tydiqa | logiqa | mmlu | hellaswag | arc_c | boolq | Average |
> |--|--|-|-|-|--|--|--|--|
> | ds2  | 41.51 ± 1.08   | 41.83 ± 2.13 | 24.6 ± 0.39  | 56.97 ± 0.14 | 55.7 ± 0.09   | 44.39 ± 0.87 | 76.99 ± 0.62 | 48.87 ± 0.12 |
> | full_token | 43.92 ± 0.52   | 43.83 ± 2.32 | 24.81 ± 0.47 | 56.95 ± 0.11 | 55.59 ± 0.04  | 44.99 ± 0.49 | 74.24 ± 0.38 | 49.17 ± 0.21 |
> | random   | 43.88 ± 0.09   | 42.77 ± 3.08 | 24.24 ± 0.74 | 57.09 ± 0.13 | 55.48 ± 0.1   | 44.96 ± 0.45 | 74.44 ± 0.53 | 48.97 ± 0.31 |
> | rho     | 47.97 ± 0.77   | 50.93 ± 0.85 | 26.15 ± 0.09 | 57.16 ± 0.01 | 56.48 ± 0.04  | 46.03 ± 0.62 | 77.04 ± 0.1  | 51.67 ± 0.15 |
> | fixed_model_cleaning| 48.24 ± 0.61   | 51.8 ± 0.47  | 26.05 ± 0.16 | 57.03 ± 0.11 | 56.46 ± 0.04  | 45.97 ± 0.51 | 77.27 ± 0.27 | 51.83 ± 0.06 |
> | self_evolving_cleaning| 52.01 ± 0.93   | 54.38 ± 0.33 | 28.43 ± 0.36 | 56.31 ± 0.12 | 55.75 ± 0.05  | 46.68 ± 0.6  | 77.25 ± 0.07 | **53.0** ± 0.2   |
>
> | Method (LLaMA-3.1-8B)  | truthfulqa  | tydiqa | logiqa | mmlu | hellaswag | arc_c | boolq  | Average|
> |--|-|--|-|-|-|-|--|-|
> | ds2     | 48.12 ± 1.3    | 53.02 ± 2.05  | 27.34 ± 0.32  | 65.81 ± 0.03   | 60.47 ± 0.1 | 53.37 ± 0.18 | 83.38 ± 0.0  | 55.93 ± 0.12 |
> | full_token   | 49.08 ± 1.36| 52.5 ± 2.83| 28.06 ± 0.47| 65.77 ± 0.02| 60.33 ± 0.08  | 54.12 ± 0.2 | 82.92 ± 0.38 | 56.1 ± 0.2   |
> | random   | 49.84 ± 1.01| 54.09 ± 1.39  | 27.7 ± 0.59   | 65.83 ± 0.06 | 60.32 ± 0.08  | 54.41 ± 0.3 | 83.24 ± 0.14 | 56.47 ± 0.12 |
> | rho | 55.76 ± 0.99| 58.49 ± 3.46  | 28.11 ± 0.93  | 65.75 ± 0.01   | 62.0 ± 0.12   | 54.95 ± 0.23  | 82.16 ± 0.45 | 58.2 ± 0.61  |
> | fixed_model_cleaning| 56.04 ± 0.11| 61.85 ± 0.72| 28.06 ± 0.27| 65.62 ± 0.08 | 61.96 ± 0.04| 55.04 ± 0.14| 82.74 ± 0.08 | 58.77 ± 0.15 |
> | self_evolving_cleaning| 59.75 ± 0.16 | 64.1 ± 0.85 | 26.56 ± 0.54| 65.17 ± 0.1| 62.65 ± 0.03  | 54.75 ± 0.2 | 82.54 ± 0.09 | **59.33** ± 0.23 |
>
> The t-test results indicate that Self-Evolving Cleaning significantly outperforms RHO, showing statistically significant improvements with a p-value less than 0.05.
>
> | Method Comparison  (T-test value, p value) | LLaMA-3.2-3B | LLaMA-3.1-8B  |
> |---|--|-|
> | Fixed-model cleaning vs. Rho| (1.77, 0.152)   | (1.56, 0.193)   |
> | Self-Evolving Cleaning vs. Rho     | (9.18, 0.001)   | (3.02, 0.039)   |
>
>
> > **Weakness 2: Computational cost**
>
> **Response**:
> Thank you for highlighting this important issue. The computational costs associated with the token cleaning pipeline primarily consist of the training costs, akin to those of standard supervised fine-tuning (SFT), along with two types of additional inference costs. These additional costs stem from one base model and another reference model, both of which are used to calculate token-level influence scores.
>
> Compared to the Rho-1 baseline, our two proposed methods do not incur any additional inference costs since the total data size for inference remains unchanged. For the Naive Fixed-Model Cleaning, we perform a one-shot inference on all samples simultaneously, mirroring the process used in Rho-1. For the Self-Evolving Cleaning pipeline, we simply segment the data pool into several partitions for independent inference using different reference models,  i.e., (50k samples -> 10k (reference model 1), 10k (reference model 2), …, 10k samples).  Consequently, the inference cost for the Self-Evolving Cleaning pipeline is also equivalent to that of Rho-1.
>
> > **Weakness 3: technical contribution clarification**
>
> **Response**:
> Thank you for raising this concern. We would like to clarify that although the concept of masking unimportant tokens, initially introduced by Rho-1, is not new but remains very timely. Based on this, our work makes several novel contributions:
> - Self-Evolving Cleaning: Our iterative approach that progressively refines token selection is entirely new and shows superior performance.
> - Theoretical Framework: We provide a rigorous analysis of when and why token cleaning outperforms full-token training, with error upper bounds that explain the observed Matthew effect.
> - Noisy-Label Perspective: We reframe token selection as a noisy-label problem, providing a new theoretical lens that connects token cleaning to a rich body of work on learning with noisy labels.
> These contributions extend beyond prior work (Rho-1) on token selection, offering both theoretical insights and practical improvements.

---

> > ### Comment · Reviewer_fypD · 2025-04-02
> >
> > Thank you for the rebuttals. Most of my concerns have been resolved. I adjusted the score accordingly.

---

> > > ### Author Response · Authors · 2025-04-03
> > >
> > > Dear Reviewer fypD,
> > >
> > > We are pleased that most of the concerns have been addressed.Thank you for taking the time to carefully consider our rebuttal and for your thoughtful feedback! We are grateful for your positive impression and support for our work.

---

### Official Review · Reviewer_wKJy · 2025-03-09

**Overall Recommendation:** 3

**Summary:**

The paper introduces a token cleaning method for supervised fine-tuning (SFT) of large language models that operates at a fine-grained, token level rather than discarding whole samples. It employs an influence-guided scoring function to assess each token’s contribution by comparing loss differences between a base model and a reference model. Tokens deemed uninformative or redundant are filtered out using a threshold, thereby preserving key task-relevant information. Two cleaning strategies are proposed: a fixed-model approach that performs one-shot cleaning and a self-evolving approach that iteratively updates the reference model. The method is supported by theoretical analysis and extensive experiments across various downstream tasks.

**Claims And Evidence:**

The authors support their claims with theoretical analysis in proving training with cleaned token is lower bounded by training with full tokens under the assumption that influence function can be used to evaluate the quality of token. The proof seems sound under the assumption of model quality. However, I do have some concerns for this assumption, which is detailed in "Other Strengths And Weaknesses." Empirical results in demonstrates self-evolving method is performing better than fixed model on Llama-3.2 and Llama 3.1 models. In general, I think the claims made in this paper are well justified.

**Essential References Not Discussed:**

N/A

**Experimental Designs Or Analyses:**

In general the proposed experiments are sound in terms of design and baseline choices.

**Methods And Evaluation Criteria:**

The proposed methods and evaluation criteria makes sense. The idea of token cleaning flows naturally from pretraining (Rho-1) to fine-tuning, and is an intuitive extension from sample-level data cleaning to token-level.

**Other Comments Or Suggestions:**

N/A

**Other Strengths And Weaknesses:**

Strength:
1. This paper has strong theoretical analysis supported in token selection during SFT, which is a novel contribution.
2. The paper is well written and easy to follow.

Weakness:
1. One concern I have is how accurate is using influence function to assess the quality of token itself. Previous work [1] indicates that influence functions might not work well on LMs, and "can lead to strong biases in this estimator." If the influence function itself is biased, the soundness of the theoretical finding in this paper might be affected. While I understand that there are differences between theoretical assumptions and practical settings, I hope the authors can provide more insights and intuitive explanations in justifying using influence functions to score the token.
2. The size of the model considered in the experiment section are 3B and 7B. I'm wondering how effective this method is when presenting to a larger and more powerful model. Will larger model more robust to the SFT noise or will it still benefit from token cleaning?

[1]Causal estimation of memorisation profiles,

**Questions For Authors:**

Refers to weakness section.

**Relation To Broader Scientific Literature:**

This work is a natural extension of Rho-1.

**Theoretical Claims:**

I did not check the proof step-by-step, but it looks promising.

---

> ### Author Rebuttal · Authors · 2025-04-01
>
> We want to thank Reviewer wKJy for their positive feedback and comments. We will address individual comments below.
>
> > **Weakness 1: how accurate is using influence function to assess the quality of token itself**
>
> **Response**: Thank you for raising this important question. We'd like to clarify with the following points:
> - The mentioned work made the claim because of the violations of required theoretical assumptions (e.g., positive-definite Hessian matrix) for language models. These assumptions are proposed for approximation. However, in this work, the method involves directly working with the difference in loss values rather than making approximations or Taylor expansion. Thus, the accuracy of the influence function would not be affected.
>
> Here is a more detailed comparison between our implementation of the influence function and the standard one who needs approximations. The standard influence form is defined as  loss($\theta$) - loss($\theta’$), given that the $\theta$ model is the current model and $\theta’$ is the model by removing a token, which requires some approximations to avoid model re-training. However, in our scenario, we do not have to counter-factually remove a token to calculate the influence since the training procedures are naturally adding tokens, which also have a form of loss($\theta$) - loss($\theta’$) but $\theta’$ can be seen as the current model (for example) and $\theta$ can be seen as the model after adding one token. Specifically, in the self-evolving clearing pipeline, we are iteratively using the influence function to identify informative tokens for the next-round training. Therefore, we can obtain those two models naturally with the original base model as $\theta$ and the current model as $\theta’$.
> Therefore, the only difference is we are adding tokens actively [1-2] while the standard approach is deleting tokens (i.e., re-training) [3].
>
> - Besides, our theoretical analysis is based on a unified and analytical framework, which mainly explains the when and why the SFT with cleaned tokens outperforms with full tokens, regardless of the specific token quality evaluation metric used. As such, our theoretical insights are independent of the precise accuracy of the influence function, focusing instead on the overall reduction in token noise rates.
>
> Finally, to address the reviewer's concern, there are more insights here.
> Intuitively, we hope that the base model, denoted as $\theta$, will perform similarly to the reference model, $ \theta' $, on downstream tasks after fine-tuning. To approximate this criterion, we use the token loss calculated by the reference model, expressed as $\\ell(x_{i,j} | x_{i,:j}, \theta') $. If the token loss calculated by the base model, $ \\ell(x_{i,j} | x_{i,:j}, \theta) $, is significantly higher than that calculated by the reference model, it indicates that this particular token requires fine-tuning to better align the base model with the reference model. Relying solely on $ \ell(x_{i,j} | \boldsymbol{x}_{i,:j}, \theta) $ as a metric for token quality—by ranking and selecting tokens based on this loss—would only consider token-level information and neglect the task-specific insights provided by the reference model.
>
> [1] Estimating Training Data Influence by Tracing Gradient Descent, NeurIPS’20.
>
> [2] Fairness without harm: an influence-guided active sampling approach, NeurIPS’24.
>
> [3] Understanding black-box predictions via influence functions, ICML’17.
>
> > **Weakness 2: larger-scale model performance evaluation**
>
> **Response**:
> Thank you for your valuable feedback. Given our constraints with GPU resources and time, we utilized LLaMA-2-13B-hf as the base model to assess the efficacy of our proposed methods. All experimental settings are consistent with the original settings. The results demonstrate the superiority of our methods.
> | Method                    | truthfulqa | tydiqa | logiqa | mmlu | hellaswag | arc_c | boolq | Average |
> |---------------------------|----------------|--------|--------|------|-----------|---------------|-------|---------|
> | base (llama-2-13b-hf)     | 36.73          | 33.79  | 26.05  | 55.14| 60.11     | 47.80         | 80.64 | 48.6    |
> | ds2                       | 37.48          | 35.47  | 27.91  | 55.19| 60.37     | 48.49         | 81.16 | 49.4    |
> | full_token                | 41.73          | 38.29  | 27.60  | 55.33| 60.33     | 49.70         | 82.40 | 50.8    |
> | random                    | 41.67          | 37.44  | 27.60  | 55.30| 60.35     | 49.61         | 82.33 | 50.6    |
> | rho                       | 44.41          | 42.12  | 28.06  | 55.35| 61.33     | 50.99         | 81.87 | 52.0    |
> | fixed_model_cleaning      | 44.03          | 40.07  | 27.91  | 55.63| 61.32     | 50.90         | 81.62 | 51.6    |
> | self_evolving_cleaning    | 49.13          | 44.65  | 26.67  | 55.42| 62.23     | 51.16         | 82.49 | **53.1**    |

---

### Official Review · Reviewer_dD3A · 2025-03-11

**Overall Recommendation:** 2

**Summary:**

This paper enhances the SFT process by scoring and filtering out uninformative tokens, treating them as noisy labels. They introduce a novel influence-guided token cleaning pipeline to address this issue. Furthermore, they offer rigorous analyses to demonstrate when and why SFT with cleaned tokens outperforms the use of the complete set of tokens.


## update after rebuttal

I still don't believe the performance improvement is due to masking out uninformative tokens. Even in the example provided by the authors, they use the phrase "negatively impactful tokens" to explain how it works. I don't think such negatively impactful tokens can be considered merely uninformative. I believe the author should distinguish between negatively impactful tokens and uninformative tokens, rather than treating them merely as generic error labels.

**Claims And Evidence:**

I question the claim to regard the selection of tokens as a noisy-label problem.

**Essential References Not Discussed:**

NA

**Experimental Designs Or Analyses:**

The evaluation seems overly simplistic. Based on my understanding, the datasets referenced pertain solely to classification tasks rather than generation tasks. If my interpretation is incorrect, I would appreciate clarification from the authors.

I suggest you compare with more baselines such as the sample-level data selection to show why the token-level one is necessary. For example, you can compare with the method in "Speculative Coreset Selection for Task-Specific Fine-tuning."

**Methods And Evaluation Criteria:**

The evaluation is relatively too easy. As far as I know, the datasets mentioned are all classification tasks instead of generation tasks. I hope the authors can correct me if my understanding is wrong.

**Other Comments Or Suggestions:**

I want to summarize my main concerns here:

1. I'm unclear on the necessity of manually selecting uninformative tokens, as this process can be naturally handled by the loss function. The loss associated with uninformative tokens can decrease naturally, serving as a form of automatic reweighting. One possible explanation is that if the loss for these tokens becomes too low, it could negatively impact generalization. Therefore, it might be necessary to prevent further reduction in their loss by adjusting their weight.

2. I find the experiments somewhat lacking, especially when considering the benchmarks and baselines used.

**Other Strengths And Weaknesses:**

The main issue is that it's not immediately clear why uninformative tokens should be considered noisy. I recommend providing a demonstration to clarify why these tokens can be categorized as such.
From my perspective, the loss associated with uninformative tokens tends to be minimal during training. In the Next-Token Prediction paradigm, the loss of uninformative tokens is naturally down-weighted due to its small value. Therefore, I don't understand the necessity of manually removing them.

**Questions For Authors:**

What is the cost of this method? I believe the cost is quite significant due to the additional inference required.

Additionally, I find the definition of the influence function somewhat unusual, though still acceptable. The primary goal should be to assess the influence of a token. I think a token's influence should be measured by the change in loss across the entire dataset before and after the token is removed. This approach differs significantly from your current definition.

**Relation To Broader Scientific Literature:**

token-level understanding in SFT

**Theoretical Claims:**

yes

---

> ### Author Rebuttal · Authors · 2025-04-01
>
> We would like to thank Reviewer dD3A for the time and effort. We will address individual comments below.
>
> > **W1: evaluation benchmarks**
>
> **Response**: The benchmarks we used are standard in SFT studies (see Related Work) and assess a broad spectrum of capabilities, including question answering, reasoning, and multilingual comprehension, not just classification. While some benchmarks use multiple-choice formats, they test the model's generative abilities and knowledge without converting it into a classification model.
>
> For your concern, we use two popular LLM-Judge benchmarks, MT-Bench and Vicuna-bench, to evaluate our model generations (the higher numbers indicate the better results). These results highlight the superiority of our methods.
> | Method (llama-3b)  | MT-Bench | Vicuna-Bench |
> |-|-|-|
> | DS2| 5.62 | 4.46|
> | Full_token| 5.40 | 4.46|
> | Random| 5.32 | 4.80|
> | Rho|6.33| 5.65|
> | Fixed-model cleaning| 6.27| 5.74|
> | Self-evolving cleaning| **6.63**| **5.76** |
>
> > **W2: Additional baseline STAFF**
>
> **Response**: We replicated the STAFF baseline with pruning rates of 40%/60%. Due to the absence of a comparable smaller model for Mistral-7B, we assessed STAFF using two LLaMA base models, using LLaMA-3.2-1B to compute speculative scores.
> While STAFF remains a strong baseline, yet Self-Evolving Cleaning shows competitive results. For instance, with LLaMA-3.1-8B, our method surpasses STAFF in average performance (59.20 vs. 57.0/56.4). Although STAFF performs better on BoolQ, Self-Evolving Cleaning consistently improves across various tasks. In our revised manuscript, we will cite this reference and discuss.
>
> | Model | TruthfulQA| TydiQA | LogiQA | MMLU | HellaSwag | ARC-C | BoolQ | Average|
> |-|-|-|-|-|-|-|-|-|
> | STAFF (llama-3b, 20k)|44.1| 45.92| 23.72| 56.86| 55.6|43.5|76.13| 49.4|
> | STAFF (llama-3b, 30k) | 43.15| 48.31| 24.03| 56.78|55.43| 44.19|73.78| 49.4|
> |Self-Evolving Cleaning (llama-3b) | 51.07| 56.38|28.22| 56.18| 55.81|45.99 |77.33 |**53.0**|
> | STAFF（llama-8b, 20k) | 46.3| 61.34| 26.82 | 65.07| 60.35| 54.61| 84.57| 57.0|
> | STAFF（llama-8b, 30k) | 46.06| 61.98 |24.03| 63.57| 60.29 | 55.12| 83.57| 56.4|
> |Self-Evolving Cleaning (llama-8b) |59.58| 63.58| 26.05 |65.07 |62.67|54.87|82.49 |**59.2**|
>
> > **W3: the necessity of manually selecting uninformative tokens**
>
> **Response**:
> We clarify the distinction between uninformative and noisy tokens as follows. In the SFT phase, every response token is labeled as 1, making it difficult to identify which tokens should be included for task-specific knowledge and which excluded for higher knowledge density. As a result, traditional labels are considered noisy, consisting of informative tokens (correctly labeled as 1) and uninformative tokens (incorrectly labeled as 1 but should be 0), with the latter corresponding to label errors.
>
> We acknowledge that while uninformative tokens have low loss, their impact on model convergence cannot be ignored. These tokens may lead the model to make trivial, non-task-specific predictions. Our findings, supported by curriculum learning literature [1], show that while "easy" patterns are learned first, "difficult" patterns require more focus. Therefore, adjusting the weights of these tokens is crucial for optimizing downstream task performance.
>
> [1] A Survey on Curriculum Learning, TPAMI'21.
>
> > **Q1: Computational cost**
>
> **Response**: There are two types of additional inference costs in our method: one associated with the base model and the other with the reference model. Besides, there is also training cost for the reference model on a small subset (10k).  Compared to the baseline Rho-1, our approach does not incur any extra inference costs. This is because the Self-Evolving Cleaning pipeline merely divides the data pool into several partitions for independent inference, i.e. (50k samples -> 10k, 10k, …, 10k samples).
>
> > **Q2: Influence function**
>
> **Response**:
> The standard influence form is defined as loss($\theta$) - loss($\theta’$), given that the $\theta$ model is the current model and $\theta’$ is the model by removing a token.
> However, in our scenario, we do not have to counter-factually remove a token to calculate the influence since the training procedures are naturally adding tokens, which also have a form of loss($\theta$) - loss($\theta’$) but $\theta’$ can be seen as the current model (for example) and $\theta$ can be seen as the model after adding one token. Specifically, Self-Evolving Cleaning iteratively use the influence function to identify informative tokens for the next-round training. Therefore, we can obtain those two models naturally with the original base model as $\theta$ and the current model as $\theta’$.
> Therefore, the only difference is we are adding tokens iteratively [1] while the standard approach is deleting tokens (re-training) [2].
>
> [1] Estimating Training Data Influence by Tracing Gradient Descent, NeurIPS’20.
>
> [2] Understanding black-box predictions via influence functions, ICML’17.

---

> > ### Comment · Reviewer_dD3A · 2025-04-03
> >
> > Thanks for clarifying the "multiple-choice formats" in the evaluation benchmarks, the comparison with additional baselines, and the computational cost.
> >
> > My comments are as follows:
> >
> > (1) The influence function in Equation (2) is not intended to evaluate the influence on the trained model caused by the token. Instead, it assesses the influence of the whole training data on the prediction of a single token $x$. This value then is used to evaluate the quality of the token $x$. I still think this is reasonable, but I believe the original manuscript's statement is unclear and potentially misleading. I do hope further clarification can be provided. However, I won’t give a negative rating because of this.
> >
> > (2) Such a definition of "influence" gave me a misleading impression that if the "influence" value p $x$ is not prominent, the token $x$ must be uninformative and potentially noisy. However, the definition of "influence" does not actually measure the influence caused by the token $x$. Thus, I don’t agree that tokens with low so-called "influence", as calculated in Equation 2, are necessarily uninformative. For example, another possibility is that these tokens may contain specialized knowledge that the model attempts to learn but inherently finds difficult—essentially, they are hard samples or knowledge that, due to their low occurrence in the training data, are neglected during the training process. While this paper doesn’t need to determine how many of these tokens are truly uninformative versus informative but difficult to learn, I hope the authors can provide some assumptions regarding the training data. For instance, the proposed method might only be effective if the proportion of informative but difficult-to-learn tokens is low. Additionally, I would appreciate a more careful discussion on the intuition behind treating such tokens as noisy. Again, this is not my main concern, and I won’t give a negative rating because of it.
> >
> > (3) My main concern remains the necessity of manually selecting uninformative tokens. While I understand that this approach is somewhat supported by existing data-centric methodologies like curriculum learning, I am trying to grasp the fundamental reason why this method is effective, especially given that only 40% of the tokens are filtered out, leaving 60%.
> > For instance, I can understand focusing solely on the top 10% of tokens—although their losses are larger on a per-token basis, the overall loss is still significantly influenced by the remaining 90% of tokens, whose losses are smaller individually but collectively impactful when accumulated in the loss function. However, in this scenario, this paper is filtering out only 40% of the tokens, whose losses are naturally small. I would hypothesize that these filtered tokens originally contribute minimally to the training loss. Why, then, do they exert such a significant influence?
> >
> > In summary, what I am trying to convey is that while I do believe the methodology could be effective under some circumstances, I don’t think it should be framed as filtering out noisy tokens. Instead, I hope to explore deeper insights into why this method works.

---

> > > ### Author Response · Authors · 2025-04-03
> > >
> > > We sincerely appreciate your thoughtful review of our responses and your valuable suggestions! We will address individual comments below.
> > >
> > > > **Concern 1**: The influence function in Equation (2) is not intended to evaluate the influence on the trained model caused by the token.
> > >
> > > **Response**: Thanks for showing the confusing point of our paper. We would like to clarify that Equation (2) is intended to evaluate the influence on **"future training data"** caused by learning with “some tokens”. Please note this definition follows the most recent novel application of influence functions [1–2]. As illustrated in Figure 1 and Algorithm 1, this generic form can serve as the score function to examine the token scores of different methods. For example, in Fixed-Model-Cleaning, the "future training data" is dataset $\widetilde D$, and “some tokens” correspond to the warmup data, meaning that learning with “some tokens” can impact the model’s predictions on different tokens differently. This difference is measured on “future training data”. If tokens in “future training data” are negatively impacted by training and we assume the model performs better after training (as explained before Eqn. (3)), we prefer to mask out these tokens in the next-step training. The same principle applies to Self-Evolving Cleaning. We’ll revise the paper and make it more clear.
> > >
> > > [1] Estimating Training Data Influence by Tracing Gradient Descent, NeurIPS’20.
> > >
> > > [2] Fairness without harm: an influence-guided active sampling approach, NeurIPS’24.
> > >
> > > > **Concern 2: additional assumption on train data**
> > >
> > > **Response**: Thank you for your thoughtful suggestion. Instead of putting assumptions on training data, we assume the model quality as in Line 188 (LHS). Specifically, we assume the model $\theta’$ performs consistently better than $\theta$ on the concerned task, then all tokens that improve the model performance should have a lower loss after training, i.e., a positive token score. Under this assumption, difficult tokens will also have a lower loss after training. We acknowledge that this assumption is strong since a consistently better model is hard to guarantee. But as long as the model is getting better in the majority cases, using our score functions to mask out tokens benefits. See more theoretical analyses in Corollary 5.2 and Sections 5.2, 5.3. In the revised version, we would more clearly highlight and discuss this assumption.
> > >
> > > > **Concern 3: the necessity of manually selecting uninformative tokens**
> > >
> > > **Response**:
> > > Thank you for your further discussion about this question. Please note our score uses the difference between token losses ($\text{loss}_1 - \text{loss}_2$) to distinguish between informative and uninformative tokens. Here, $\text{loss}_1$ represents the token loss on the base model, and $\text{loss}_2$ is the token loss on the reference model.
> > > We would like to clarify this in the following ways:
> > > 1. **loss $\\to 0$ is only a necessary condition of score $\\to 0$**. For example, we will have the same token score of 0.1 when $\text{loss}_1 - \text{loss}_2 = 0.2 - 0.1$ and $\text{loss}_1 - \text{loss}_2 = 4.2 - 4.1$, respectively. So our selection mechanism does not mask out all the difficult tokens.
> > > 2. **The sign of our scores matters.** For example, a difficult token may have fluctuating loss $\text{loss}_1 = 2$ and $\text{loss}_2 = 4$, or $\text{loss}_1 = 4$ and $\text{loss}_2 = 2$. The former case can be easily captured by our method when the threshold is 40% of low-score tokens since the score is negative. Therefore, our method is not simply masking out the tokens whose absolute scores $\\to 0$. We tend to mask out tokens that have a negative impact on model training. In other words, uninformative tokens are task-irrelevant (as mentioned in Section 4.1). They may cause a negative impact (should be masked out) or minimal impact (prefer to be masked out according to the logic of curriculum learning).
> > > 3. **Masking out 90% vs. 40%**. As illustrated in Figure 2, 40% works well empirically. It is also strategically reasonable. Let's consider the following motivating example. If we have a data sample with 10 tokens, where half are positively impactful and half are negatively impactful, our methods would involve filtering out the 4 most negatively impactful tokens. While 40% may not represent a 'perfect' threshold for identifying detrimental tokens, it is a more fault-tolerant number since even the scores made some mistakes, keeping some tokens whose scores are close to zero could be a better solution than aggressively removing 90% tokens.

---

### Official Review · Reviewer_W8kk · 2025-03-14

**Overall Recommendation:** 4

**Summary:**

Recent large language models are trained in multiple stages with vast amounts of data to achieve stellar performance. However such a training regime also brings in several challenges, including the challenge of observing similar high frequency phrases over and over again. This work aims to improve the supervised fine-tuning of large language models by performing token cleaning, masking out uninformative tokens while preserving the important ones. Through theoretical and experimental analyses, the work further aims to highlight the strengths of the proposed token cleaning pipeline.

**Claims And Evidence:**

The paper is built on the intuition that during SFT, the model observe a reiteration of the previously learned concepts in addition to the new information that the SFT aims to distill. The authors formulate this challenge analogously with a noisy label scenario and take inspiration from that literature. Specifically, they utilize influence functions and a hard thresholding mechanism on top for _cleaning the tokens_ that are not very informative. The authors then claim that this approach should _improve the optimization process by preventing misleading gradients_.

With respect to this claim, the authors present theoretical analyses in addition to experimental results on several well-established large language models. While the theoretical analyses is not fully precise, e.g in Sections 5.1 & 5.2 it makes a few trivial observations such as the challenges caused by the noise in tokens, I believe that that the overall discussion and the examples used (especially the discussion in Section 5.3) are valuable for highlighting the motivations behind the work and support the claims.

In addition, as shown in Table 1, the proposed approach(es) mostly improves the performance of different well-established models on several benchmarks. Thus, the overall claims of the paper are well-supported.

**Essential References Not Discussed:**

N/A

**Experimental Designs Or Analyses:**

The general experimental design follows the established standards and the analyses are sound based on the quantitative evidence.

**Methods And Evaluation Criteria:**

- Benchmarks are well-known datasets in the field and are sufficient to support the claims and highlight the contributions of the work.
- Utilized baseline models (namely two exemplars from the LLaMA-3 Herd and Mistral) are also reasonable and sufficient

**Other Comments Or Suggestions:**

N/A

**Other Strengths And Weaknesses:**

Overall, the presented figures, tables and the algorithm are neatly explained. The narrative of the work is also fluent and was not very challenging to read through. I believe that this work overall is a strong and meticulous submission with interesting and well-supported ideas presented in it.

**Questions For Authors:**

- How do you think token cleaning effects the supervised fine-tuning efficiency, both with respect to the data requirements and number of training iterations?

**Relation To Broader Scientific Literature:**

Supervised fine-tuning is a widely adopted approach in large language model training. The provided issue in the work, namely the reiteration of the same knowledge, is also a very real problem, especially considering the scale of the pretraining data these models go through. Based on this, the work proposes an interesting and well-founded approach to addressing this issue and providing a more efficient training. I believe that the work could be interesting to the broader community.

**Theoretical Claims:**

I checked the theoretical claims and could not find any obvious issues. However, I would like to note that the theoretical claims do not explicitly support the proposed approach: They mainly provide a more intuitive explanation as to why the proposed approaches could ultimately work under several strong assumptions. I do not believe that this casts a shadow with respect to the support of the claims, but rather that the theoretical part is a more minor factor in doing so.

---

> ### Author Rebuttal · Authors · 2025-04-01
>
> We want to thank reviewer W8kk for the positive feedback! We will address individual comments below.
> > While the theoretical analyses is not fully precise, e.g in Sections 5.1 & 5.2 it makes a few trivial observations such as the challenges caused by the noise in tokens...
>
> **Response**: We agree that LLM learning dynamics and convergence are much more complicated than our analytical framework, but our analyses in Section 5, form a complete theoretical framework based on error upper bounds that provides valuable support for our claims.
> Our main claim is that the two proposed token cleaning methods can effectively improve LLM SFT performance. Our theoretical analyses support this claim in two important ways:
>
>  **Explaining why the proposed methods work**:
> - Theorem 5.1 and Corollary 5.2 establish an analytical framework that precisely describes when token cleaning outperforms using all tokens through the trade-off between data quality (η) and data quantity (M);
> - Section 5.2 analyzes the characteristics of Fixed-Model Cleaning, explaining why this method provides stable but bounded improvements
> - Section 5.3 analyzes the dynamic behavior of Self-Evolving Cleaning through the Matthew Effect, explaining why this method can produce more significant improvements in certain scenarios
>
>  **Provide practical guidance for method selection and application**:
>  - Clearly identifying when Self-Evolving methods might lead to performance degradation (G2 group data)
> - Providing theoretical basis for selecting appropriate cleaning methods in different scenarios
> - These alerts and guidance are validated in our experiments, as seen in Table 2 where performance changes across different tasks align well with theoretical predictions
>
> We agree that while some analyses provide foundational observations, our theoretical framework offers deep insights into method behaviors, supporting our claims and guiding practical applications. In the revised paper, we will clarify the links between theory and experiments to explicitly highlight the importance of the theoretical aspects.
>
>
> > However, I would like to note that the theoretical claims do not explicitly support the proposed approach...
>
>  **Response**: We agree that our theoretical analyses primarily serve as intuitive explanations rather than explicit proofs for our proposed approaches, and they are indeed based on certain simplifying assumptions.
> We acknowledge that the experimental results constitute the primary evidence supporting our claims about the effectiveness of our token cleaning methods. The theoretical analyses were intended to complement these empirical findings by:
> - Providing a conceptual framework to understand when and why token cleaning can be beneficial
> - Offering explanations for the observed performance patterns across different methods
> - Helping practitioners make informed decisions about method selection in various scenarios
>
> We appreciate the reviewer's perspective that this does not diminish the overall support for our claims. In the revised manuscript, we will clarify the role of the theoretical analyses as providing intuitive explanations and insights, rather than positioning them as rigorous proofs of our methods' effectiveness.
>
> > How do you think token cleaning effects the supervised fine-tuning efficiency, both with respect to the data requirements and number of training iterations?
>
> **Response**: Token cleaning affects supervised fine-tuning efficiency in several important ways:
> - Regarding data efficiency, our approach selects only 60% of the most informative tokens rather than utilizing all tokens. This effectively reduces the data requirement by focusing the model's learning on tokens that contribute most significantly to performance. For our self-evolving cleaning method, we divide the total data samples into several chunks, each representing one iteration phase, which allows for progressive refinement of token selection.
> - Despite this data partitioning for self-evolving cleaning, the number of training iterations remains consistent with standard SFT. Therefore, token cleaning does not impact the efficiency of SFT in terms of training iterations.
>
> However, it's important to note that compared to standard SFT, token cleaning does not achieve computational speedup because the next-token prediction $x_{ij}$ still relies on the context formed by previous tokens $x_{i,:j}$. The current implementation simply masks out uninformative tokens to ignore their token loss, which is simple and compatible with all training paradigms.
> Existing token-level approaches, including RHO and our methods, mainly focus on performance efficiency rather than data (token) training efficiency. In practice, the GPU memory usage of these methods remains the same as when using full tokens. Investigating and enhancing token training efficiency represents a promising direction for future research, such as exploring ways to skip uninformative token memory occupation altogether.

---

> > ### Comment · Reviewer_W8kk · 2025-04-03
> >
> > I appreciate the detailed clarifications provided by the authors both to my review and to other reviewers. While I also agree with Reviewer dD3A on exploring why this works in more detail could be valuable, I do not think it undermines the existing contributions of the work. Thus, I would like to maintain my positive rating.

---

> > > ### Author Response · Authors · 2025-04-04
> > >
> > > Dear Reviewer W8kk,
> > >
> > > Thank you so much for taking the great effort to review our paper as well as other reviewers' comments.  We sincerely appreciate your positive impression and support for our work.  Your encouraging comments mean a lot to us. Wishing you all the best in your professional and personal endeavors!
> > >
> > > Authors

---

### Decision · Program_Chairs · 2025-05-01

**Decision:**

Accept (poster)

**Comment:**

A summary of the strengths and weaknesses based on the reviews and rebuttal (including the follow-up discussion among the reviewers and authors) is provided below.


**STRENGTHS**

1\ The authors introduced a new influence-guided token cleaning pipeline in supervised fine-tuning of large language models (instead of discarding whole samples).

2\ The proposed approach generally performs better empirically under different LLMs and benchmarks. This is further demonstrated with the additional results in the rebuttal.

3\ The theoretical analyses provide intuitive explanations and insights.


**WEAKNESSES**

1\ *Clarity*: A major concern is that the authors describe uninformative tokens as either noisy/negatively impactful or minimally impactful. I agree with Reviewer dD3A that this can be extremely confusing to a reader. The community would generally consider uninformative to mean not containing information. Does negative impact then imply no information?

To eliminate ambiguity and confusion, I would strongly recommend that the authors avoid using the words "informative" or "uninformative", and simply revert to using positively impactful, negatively impactful or minimally impactful. We trust that the authors will do the necessary to resolve this major yet rectifiable issue of clarity. You can also consider using words/definitions like "helpful" vs. "harmful", like in the following missing reference which the authors can consider citing:

Helpful or Harmful Data? Fine-tuning-free Shapley Attribution for Explaining Language Model Predictions. ICML 2024.

Secondly, among the tokens that are filtered out, what proportion of the 40% corresponds to negatively impactful and what proportion of it corresponds to minimally impactful? The authors should show these empirical results in their revised paper.


2\ *Time efficiency* of the proposed method should be analyzed rigorously (Reviewers dD3A, fypD).


From the above, the pros outweigh the cons.

The authors are strongly encouraged to revise their paper based on the above feedback and that of the reviewers.